# Genetic variance in the murine defensin locus modulates glucose homeostasis

Stewart W C Masson [1,2✉], Rebecca C Simpson[2,3,4], Harry B Cutler [1,2], Patrick W Carlos[5,6], Oana C Marian [2,7], Belinda Yau [2,7], Meg Potter[1,2], Søren Madsen [1,2], Kristen C Cooke[1,2], Niamh R Craw[1,2], Oliver K Fuller [1,2], Dylan J Harney [1,2], Mark Larance [2,7], Gregory J Cooney [2], Grant Morahan[8], Erin R Shanahan[1,2], Melkam A Kebede [2,7], Christopher Hodgkins [1,2], Richard J Payne[5,6], Jacqueline Stöckli[1,2] & David E James [1,2,7✉]

## Abstract

**Insulin resistance is a heritable risk factor for many chronic diseases; however, the genetic drivers remain elusive. In seeking these, we performed genetic mapping of insulin sensitivity in 670 chow-fed Diversity Outbred in Australia (DOz) mice and identified a genome-wide significant locus (QTL) on chromosome 8 encompassing 17 defensin genes. By taking a systems genetics approach, we identified alpha-defensin 26 (Defa26) as the causal gene in this region. To validate these findings, we synthesized Defa26 and performed diet supplementation experiments in two mouse strains with distinct endogenous Defa26 expression levels. In the strain with relatively lower endogenous expression (C57BL/6J) supplementation improved insulin sensitivity and reduced gut permeability, while in the strain with higher endogenous expression (A/J) it caused hypoinsulinemia, glucose intolerance and muscle wasting. Based on gut microbiome and plasma bile acid profiling this appeared to be the result of disrupted microbial bile acid metabolism. These data illustrate the danger of single strain over-reliance and provide the first evidence of a link between host-genetics and insulin sensitivity which is mediated by the microbiome.**

**Keywords** Insulin Sensitivity; Diversity Outbred; Microbiome; Defensin; Bile Acids
**Subject Categories** Genetics, Gene Therapy & Genetic Disease; Metabolism

## Introduction

Insulin is among the most potent hormones in the human body; therefore, it is not surprising that defects in insulin action, such as insulin resistance (IR) are shared risk factors for many chronic diseases (James et al, 2021). Studies in twins and first-degree relatives of individuals with Type 2 Diabetes have provided strong evidence of a genetic component (Poulsen et al, 2005; Warram et al, 1990). However, with some notable exceptions (Parks et al, 2015; Scott et al, 2012; Williamson et al, 2023), genome-wide association studies have failed to identify loci for IR. One potential explanation is that the diversity of human environments confounds genetic signals via complex gene-by-environment interactions. One manifestation of the environment that has been implicated in metabolic disease is the gut microbiome (Ghorbani et al, 2021; Liu et al, 2022; Takeuchi et al, 2023a). Intriguingly, some microbial compositions have been shown to have beneficial effects on metabolic health while others appear harmful (Kau et al, 2011). These effects are thought to be conveyed via diverse signalling molecules including peptides (Plovier et al, 2017; Yoon et al, 2021), metabolites (Canfora et al, 2019; De Vadder et al, 2016) and bile acids (Kreznar et al, 2017; Qiu et al, 2021; Wahlström et al, 2016; Zhao et al, 2024). However, the host-microbiome relationship is bi-directional, as host genetics can also influence gut microbial composition. Compelling evidence of this has been provided by human genome-wide association and twin studies (Goodrich et al, 2016; Lopera-Maya et al, 2022). Furthermore, a recent study in mice revealed that one quarter of all detected microbial taxa have significant quantitative trait loci (QTL) implying host regulation of their abundance (Zhang et al, 2023). Notably, this includes the metabolically beneficial *Akkermansia muciniphila*. However, it is unknown if genetic drivers of microbe abundance can modulate insulin sensitivity.

One mechanism for host regulation of the gut microbiota is the secretion of antimicrobial peptides called defensins into the intestinal lumen. Defensins are an ancient component of the

[1]School of Life and Environmental Sciences, Faculty of Science, The University of Sydney, Sydney, NSW, Australia. [2]Charles Perkins Centre, The University of Sydney, Sydney, NSW, Australia. [3]Melanoma Institute Australia, Sydney, NSW, Australia. [4]Sydney Medical School, Faculty of Medicine and Health, The University of Sydney, Sydney, NSW, Australia. [5]School of Chemistry, Faculty of Science, The University of Sydney, Sydney, NSW, Australia. [6]Australian Research Council Centre of Excellence for Innovations in Peptide and Protein Science, Sydney, NSW, Australia. [7]School of Medical Sciences, Faculty of Medicine and Health, The University of Sydney, Sydney, NSW, Australia. [8]Centre for Diabetes Research, Harry Perkins Institute of Medical Research, Murdoch, WA, Australia. ✉E-mail: stewart.masson@sydney.edu.au; david.james@sydney.edu.au

immune system, found across the tree-of-life from plants to humans. They are small peptides (29–40 amino acids) with potent antibacterial and antiviral activities (Ganz, 2003). In humans, α-defensins are produced by neutrophils and specialised gut epithelial cells, called Paneth cells. However, in mice, Paneth cells are the only source of α-defensins (also called cryptidins) (Wilson et al, 2013). Defensin secretion is regulated in response to bacteria, nutrients or cholinergic agonists (Ouellette and Selsted, 1996). Genetic variation in α-defensin expression among different mouse strains has been reported (Gulati et al, 2012) although this has not been systematically examined. Defensins have also previously been linked to metabolic health. Oral administration of human α-defensin-5 to mice ameliorated liver fibrosis, diet-induced non-alcoholic liver steatosis, dyslipidaemia and glucose intolerance (Larsen et al, 2019; Li et al, 2020; Nakamura et al, 2023; Oh et al, 2015). There has been some interest in defensins as a therapeutic due to their oral bioavailability, a characteristic lacking in most peptide-based therapies.

Mice are a valuable tool for studying human diseases due to their genetic and physiologic similarities. Moreover, they afford the precise environmental control required to detect genetic loci associated with complex diseases. This has been illustrated by studies in panels of inbred mouse strains fed different diets (Bachmann et al, 2022; Benegiamo et al, 2023; Karimkhanloo et al, 2023; Nelson et al, 2022; van Gerwen et al, 2024) and genetically diverse mouse populations such as Jackson Laboratory's Diversity Outbred (DO) mice (Chesler et al, 2016; Churchill et al, 2012; Gatti et al, 2014; Svenson et al, 2012). We have established a similar population of mice, which we term Diversity Outbred in Australia (DOz), and have previously used this population to investigate skeletal muscle insulin resistance (Masson et al, 2023b), and the metabolic consequences of weight-cycling (Thillainadesan et al, 2024). Here, we use these mice to explore genetic drivers of insulin sensitivity. Using the Matsuda Index as a surrogate measure of whole-body insulin sensitivity, we identify two quantitative trait loci (QTL), one within Ptprt, and the other within the defensin locus. Sequence variation at the defensin locus were associated with increased expression of α-defensin 26 (Defa26) and the abundance of metabolically beneficial microbes. We validated this observation via dietary supplementation with synthetic Defa26 in C57BL/6J mice, before going on to demonstrate maladaptive effects in another strain, potentially via changes in microbial derived bile acids. These data provide insight into the genetic architecture of insulin sensitivity and highlight the limitations of testing therapeutics in single mouse strains.

# Results

## Whole body insulin sensitivity is genetically linked to the defensin locus on chromosome 8

To quantify whole body insulin sensitivity, we conducted glucose tolerance tests on 670 chow-fed male DOz mice (Fig. 1A) and used these data to calculate the Matsuda Index, a surrogate measure of insulin sensitivity (Matsuda and DeFronzo, 1999). Consistent with our previous work (Masson et al, 2023b), we observed profound variation in the Matsuda Index despite all animals being housed under identical conditions and fed the same chow diet. We next

performed genetic mapping of the Matsuda Index (Fig. 1B) and identified 2 genome-wide significant quantitative trait loci (QTL), one on chromosome 2 at 162.2 Mbp and a second on chromosome 8 at 21.7 MBp (Fig. 1B). The QTL on chromosome 2 centred within Ptprt, the gene encoding receptor-type tyrosine-protein phosphatase T. Interestingly, whole body knockout of Ptprt is protective against high-fat diet induced insulin resistance (Feng et al, 2014), and receptor-type tyrosine-protein phosphatase T has been shown to regulate the stability of catenin proteins (Wang et al, 2019), several of which have been linked to glucose homeostasis (Dissanayake et al, 2018; Dissanayake et al, 2020; Masson et al, 2023a; Masson et al, 2020; Rizwan et al, 2024; Sorrenson et al, 2016). The chromosome 8 locus centred over the defensin gene cluster, a syntenic region shared between mice and humans (Patil et al, 2005), which contains 53 defensin genes and 22 defensin pseudogenes (Fig. 1B). In mice, defensins are secreted from Paneth cells in the intestinal crypt into the gut lumen to modulate microbial composition (Fig. 1C). Based on previous work (Larsen et al, 2019; Li et al, 2020; Nakamura et al, 2023; Oh et al, 2015), the link between the microbiome and metabolic health, and on-going interest in gut-derived peptides as therapeutics for metabolic disease, we selected this QTL for further validation.

## A single nucleotide polymorphism within the defensin locus associates with insulin sensitivity and Akkermansia muciniphila abundance

We conducted single nucleotide polymorphism (SNP) analysis of the defensin locus and identified several subthreshold SNPs, as well as one SNP (rs23754102) and one structural variant (SV_8_21749161_21749163) with genome-wide significant logarithm of the odds (LOD) scores (Fig. 2A). To stratify mice for further analysis we selected rs23754102, and categorised mice by the number of minor alleles they carried (AA = no minor alleles, AB = one copy, BB = two copies). Animals carrying the AB allele had identical glucose tolerance to 'control' AA mice despite a 50% reduction in insulin at the 15-min time point (Fig. 2B,C) and increased insulin sensitivity determined by the Matsuda Index (Fig. 1D). While we cannot make claims about the causal SNP, these data suggest genetic variance at the defensin locus is linked to improved whole body insulin sensitivity.

Based on defensins role in modulating the microbiome (Ganz, 2003; Ouellette and Selsted, 1996), we performed 16S rRNA sequencing on the caecal microbiomes of three groups: (1) mice carrying the putative protective rs23754102 allele (AB), (2) cage mates of these mice, and (3) a subset of AA control mice that neither carry the protective rs23754102 allele, nor share a cage with a mouse that does. Because mice are coprophagic and cage mates are distantly related, not littermates as are common in inbred strains, we can compare insulin sensitivity and microbial composition between cage mates and AA control mice. Evidence of increased insulin sensitivity or increased abundance of metabolically beneficial microbes in cage mates of AB would support a microbiome mediated mechanism of action.

All AB mice had been separately housed, and cage mates were selected at random from each of these 5 cages. AA control mice were chosen at random from across 7 different cages that had no AB mice. All three groups had equivalent alpha-diversity (Fig. 2E) and on average, the cage mates of AB mice trended towards greater

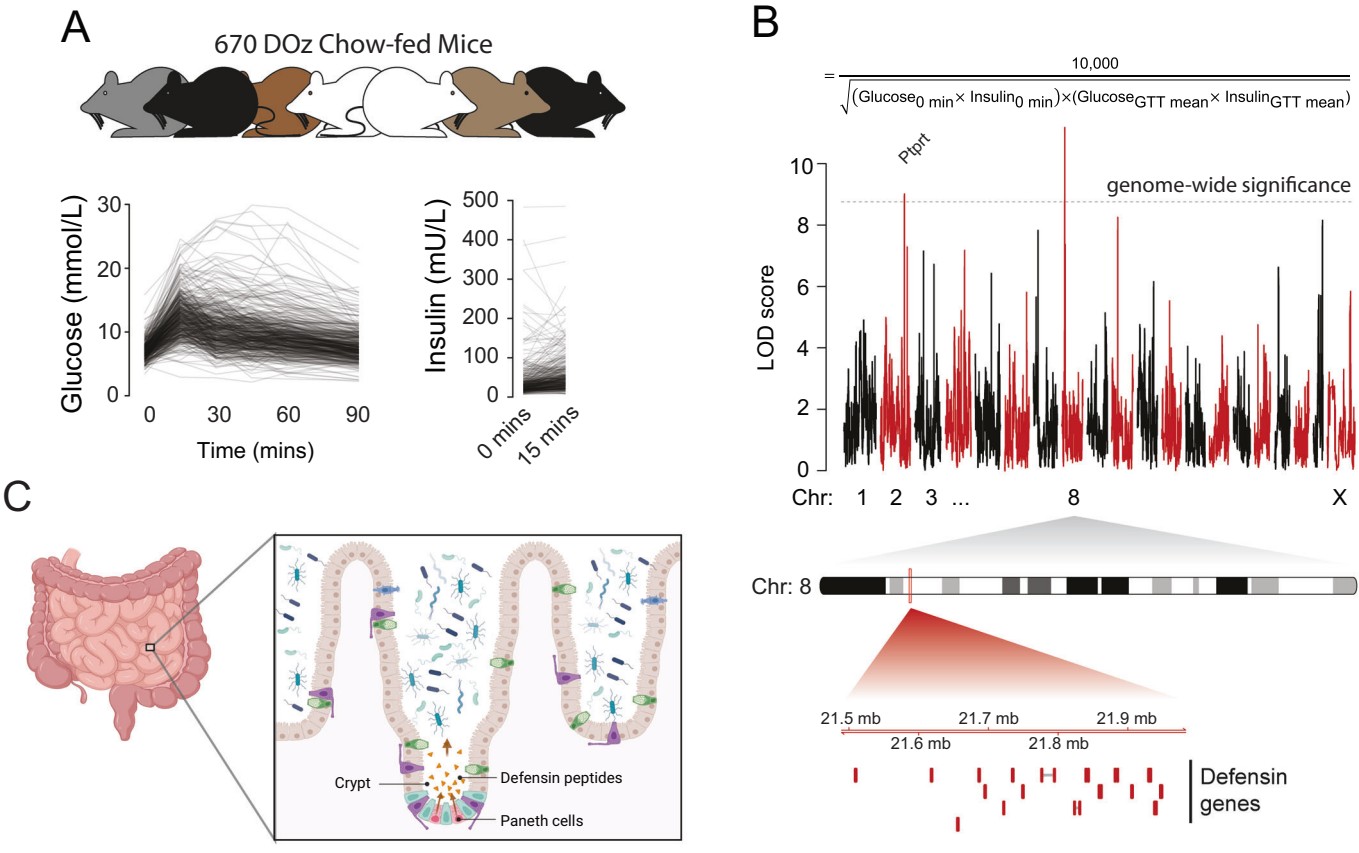

**Figure 1. Genetic mapping of insulin sensitivity in 670 chow-fed Diversity Outbred in Australia mice.**

(A) Glucose and insulin concentrations during a glucose tolerance test in 670 DOz mice. (B) The equation used to calculate the Matsuda Index and genetic mapping of the Matsuda Index in DOz mice, including schematic of the defensin locus. (C) Schematic of murine defensin secretion from Paneth cells into the small intestine. Biological replicates are shown as individual data points. Source data are available online for this figure.

insulin sensitivity than AA mice (Fig. 2C, $P = 0.09$). Beta-diversity analysis (Bray-Curtis dissimilarity and PERMANOVA) (Fig. 2F) of the three groups revealed a trend towards differences in overall microbiome composition between AB mice and AA controls ($P$ value = 0.05), and between AA controls and cage mates of AB mice ($P$ value = 0.055), but not between AB mice and their cage mates ($P$ value = 0.622). Analysis of the combined groups (AB mice + cage mates), against the AA control mice revealed divergent microbiomes ($P$ value = 0.009).

Differential abundance analysis of these groups by Analysis of Compositions of Microbiomes with Bias Correction (ANCOM-BC) revealed that compared to AA controls, *Akkermansia muciniphila* (Fig. 2G), *Bifidobacterium pseudolongum* (Fig. 2H) and *Ligilactobacillus spp.* (Fig. 2I), were higher in cage mate and AB mice, while *Limosilactobacilius spp.* (Fig. 2J) were lower. Eleven taxa were differentially abundant between AB mice and their cage mates (Appendix Fig. S1A), excluding those which are not different between AA and AB mice leaves 4 microbes with abundance patterns consistent with microbial transfer (Appendix Fig. S1B–E). *Blautia*, *Osciilospira*, and *Lachnospiraceae-45410* were lower in Cage mates than AA mice, and lower again in AB mice. While Family_Erysipelatoclostridiaceae was enriched in AB mice relative to both AA and Cage mates. Out of all differentially abundant

microbes, *A. muciniphila* stood out with strong links to metabolic health and has been shown to increase in response to α-defensin administration in C57BL/6J mice (Depommier et al, 2019; Larsen et al, 2019; Li et al, 2023; Plovier et al, 2017; Yoon et al, 2021). These data are consistent with AB mice possessing altered microbiomes relative to AA mice, and this can be transmitted to their cage mates.

## Alpha-defensin 26 positively correlates with insulin sensitivity and with founder strain contributions towards the Matsuda QTL in the defensin locus

We next sought to identify the specific defensin isoform that confers increased insulin sensitivity. To do this we quantified small intestine defensin isoform protein expression and insulin sensitivity in the Diversity Outbred founder strains (Fig. 3A). One advantage of genetic analyses in DOz is that the QTL analysis also provides the contribution of each founder strain towards the QTL signal, and this allows validation experiments to be conducted in the founder strains. Consistent with our previous work (Nelson et al, 2022) and that of others (Bachmann et al, 2022; Benegiamo et al, 2023), we observed significant variation in glucose tolerance (Appendix Fig. S2A–H), insulin sensitivity, and adiposity between the inbred

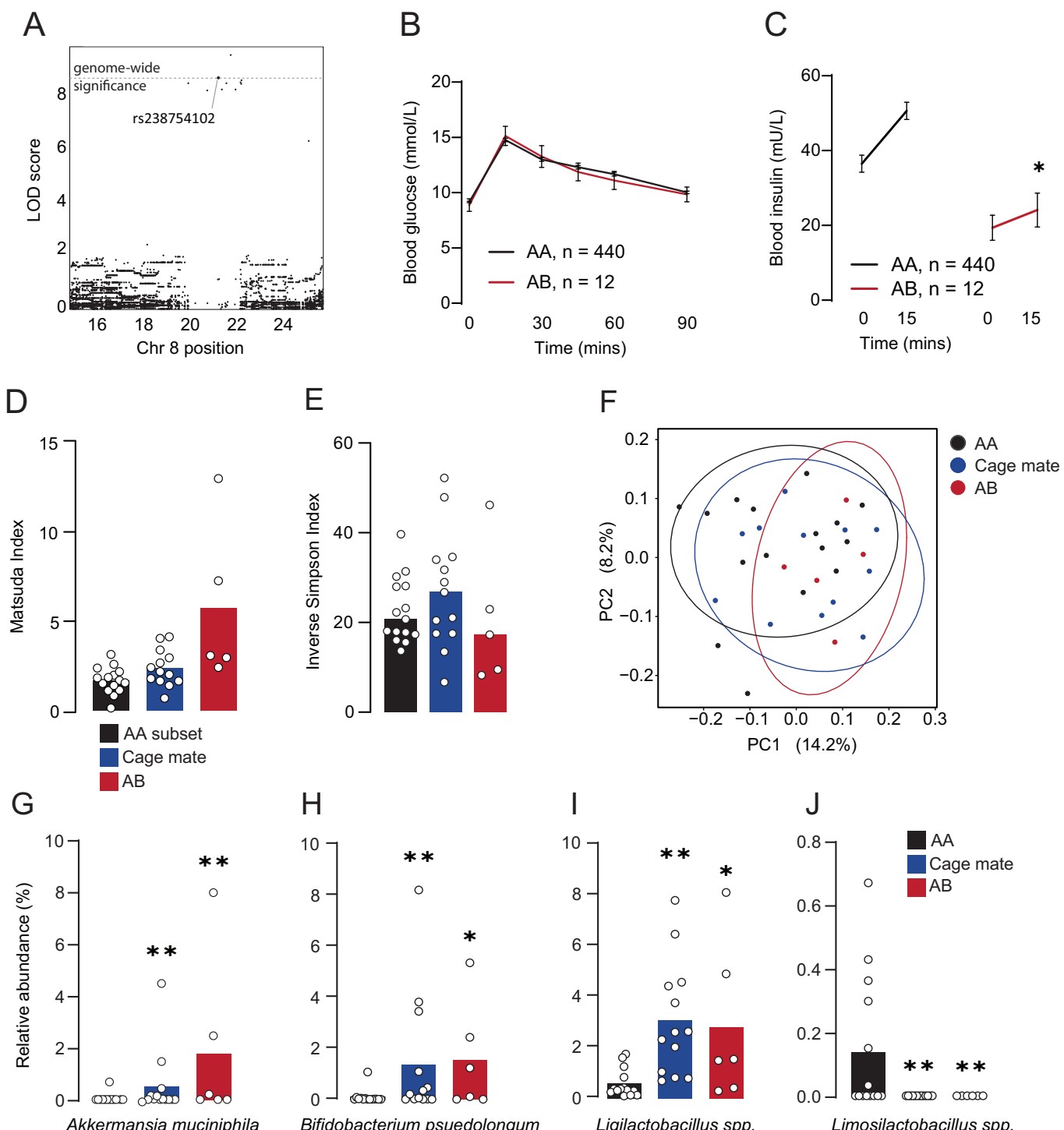

founder strains (Fig. 3B,C). Using liquid chromatography coupled to tandem mass spectrometry we detected 9 distinct defensin isoforms across 7 strains: C57BL/6J, A/J, NOD/ShiLtJ, 29S1/SvlmJ, CAST/EiJ, NZO/HILtJ and WSB/EiJ. To rule out differences in Paneth cell abundance (Gulati et al, 2012) influencing the apparent defensin expression levels, we normalised defensin peptide expression to the average of all defensins within each strain. Correlation analysis of normalised defensin isoform expression revealed that a single isoform, α-defensin 26 (Defa26), was positively correlated with insulin sensitivity ($R = 0.73$, $P < 0.05$; Fig. 3D,E). We then calculated the QTL founder effects for Matsuda Index on the chromosome 8 locus: A/J, NOD/ShiLtJ, and to a lesser extent 129S1/SvlmJ, contributed positively while CAST/EiJ, NZO/HILtJ, WSB/EiJ, C57BL/6J and PWK/PhJ contributed negatively towards the QTL (Fig. 3F). Correlating the founder effects at 21.7 MBp on chromosome 8 with mean normalised defensin levels revealed that

**Figure 2.   Analysis of microbial composition and co-housing effects in mice carrying a putative insulin sensitivity allele within the defensin locus.**

(A) Single nucleotide polymorphism (SNP) mapping of the Matsuda Index on chromosome 8. (B) Blood glucose concentrations during a glucose tolerance test of mice carrying the minor allele of rs23754102 (AB) and non-carrier (AA) controls. (C) Blood insulin concentrations during a glucose tolerance test of mice carrying the minor allele of rs23754102 (AB) and non-carrier (AA) controls. (D) Insulin sensitivity (Matsuda Index) of mice carrying the minor allele of rs23754102 (AB), non-carrier (AA) controls, and cage mates of AB mice. (E) Alpha-diversity (Inverse Simpson) of microbiomes from AA, Cage mate, and AB mice. (F) PCoA visualisation of beta-diversity between AA, Cage mate, and AB mice calculated by Bray-Curtis dissimilarity. (G) Relative abundance of Akkermansia muciniphila, (H) Bifidobacterium pseudolongum, (I) Ligilactobacillus spp. (J) Limosilactobacillus spp. in AA, Cage mate, and AB mice. Data are mean with biological replicates shown as individual data points. **$P < 0.01$, *$P < 0.05$ compared to AA mice. Metabolic phenotypes were compared using two-way RM ANOVA, with Tukey's LSD (B, C), one-way ANOVA (D). Alpha-diversity was compared by one-way ANOVA (E). Difference in microbial relative abundance (G–J) was identified by Analysis of Compositions of Microbiomes with Bias Correction (ANCOM-BC). Source data are available online for this figure.

only alpha-defensin 26 levels varied in accordance with the contributions of each founder strain to the QTL ($R = 0.89$, $P < 0.01$; Fig. 3G). This analysis is likely underpowered. Previous work has highlighted relationships between alternative defensin isoforms and metabolic health (Larsen et al, 2019; Oh et al, 2015; Puértolas-Balint and Schroeder, 2023), so with a larger panel of mouse strains it would be possible to identify other defensin isoforms which associate with insulin sensitivity.

## Alpha-defensin 26 dietary supplementation improves insulin sensitivity in HFD-fed C57BL/6J mice

Genetic mapping in DOz mice and analysis of the founder strains revealed that alpha-defensin 26 is a positive regulator of insulin sensitivity. To explore whether alpha-defensin 26 could protect against diet-induced insulin resistance, we undertook dietary supplementation studies by synthesising the lumenal (to mimic post-processing secretion) form of alpha-defensin 26 by 9-fluorenylmethyloxycarbonyl-solid-phase peptide synthesis (Fmoc-SPPS), followed by folding (Franck et al, 2020). As a positive control, we synthesised the lumenal form of human alpha-defensin 5 as this peptide has previously been shown to improve glucoregulatory control in C57BL/6J mice (Larsen et al, 2019). Consistent with previous work (Larsen et al, 2019), mice fed a western diet (WD) supplemented with Fmoc-SPPS synthesised alpha-defensin 5 (Defa5) had attenuated weight gain and reduced adiposity but normal lean mass (Appendix Fig. S3A–C) when compared to control mice fed a control WD. Defa5 supplementation also improved insulin sensitivity, evidenced by equivalent glucose tolerance but lower insulin levels relative to control mice (Appendix Fig. S3D,E). These results indicate that Fmoc-SPPS synthesised alpha-defensin peptides behave comparably to previous peptides generated by traditional expression systems (Larsen et al, 2019; Li et al, 2020; Nakamura et al, 2023). Based on these results, we proceeded with alpha-defensin 26 supplementation (Defa26).

Male C57BL/6J mice were fed either a control WD or WD supplemented with alpha-defensin 26 (WD + Defa26) for 8 weeks. On average WD + Defa26 mice gained less overall body mass and adipose tissue (Fig. 4A,B), but had equivalent lean mass relative to control WD fed mice (Fig. 4C). This reduction in adipose tissue does not appear to be the result of reduced food intake as this was comparable between groups (Fig. 4D). We also observed no difference in specific tissue weights (Fig. 4E). Although both groups exhibited near identical glucose tolerance (Fig. 4F), WD + Defa26 fed mice had lower circulating levels of insulin at the 15 min timepoint of a GTT (Fig. 4G) and higher Matsuda Index (Fig. 4H)

suggesting improved insulin sensitivity. To test this further we conducted insulin tolerance tests and measured insulin secretion in ex vivo islets in WD, and WD + Defa26 fed C57BL6/J mice (Fig. 4I). We observed a greater suppression of glucose 20–30 min after intraperitoneal insulin injection in mice supplemented with defensin relative to WD fed controls but no difference in basal or glucose-stimulated insulin secretion, or islet insulin content (Fig. 4J,K). This strongly suggests Defa26 supplementation increases insulin sensitivity in C57BL6/J mice.

In an attempt to profile potential mechanisms underpinning improved insulin sensitivity in WD + Defa26 mice we measured gut permeability by measuring FITC fluorescence in plasma, following oral gavage of FITC-dextran (Fig. 4L). Consistent with reduced gut permeability, WD + Defa26 fed mice had lower fluorescence than WD fed controls. Considering that genetic variance in the defensin locus associated with increased abundance of certain metabolically beneficial microbes we carried out 16S rRNA sequencing of caecal contents from WD and WD + Defa26 fed mice (Fig. 4M). In validation of associations between defensin locus SNPs and microbial taxa, caecums from WD + Defa26 fed mice were enriched for *A. muciniphila* and depleted of *Limosilactobacillus spp*. We also detected increased *Alloprevotella spp*. a microbe whose abundance has previously been linked to defensin supplementation (Larsen et al, 2019).

These results suggested that Defa26 supplementation could protect against WD induced insulin resistance potentially via reduced adiposity and improved gut barrier integrity. Furthermore, caecum microbial composition of WD + Defa26 fed mice exhibits similar changes than the ones observed in mice harbouring a putative protective SNP within the defensin locus.

## Alpha-defensin 26 dietary supplementation induces hypoinsulinemia, glucose intolerance and muscle wasting in WD-fed A/J mice

In view of the responses observed in C57BL/6J mice, we next performed experiments in A/J mice. A/J mice are protected from diet-induced insulin resistance (Nelson et al, 2022; Surwit et al, 1995) and express relatively high levels of Defa26 (Fig. 3E), and so we hypothesised that dietary supplementation in this strain would have no effect on whole-body metabolism. Consistent with this hypothesis and unlike C57BL/6J mice, A/J mice fed WD + Defa26 exhibited comparable weight gain (Fig. 5A) and adiposity (Fig. 5B) to WD fed controls. Surprisingly however, A/J mice supplemented with Defa26 exhibited a striking (~1 g) reduction in lean mass relative to WD fed controls (Fig. 5C). This decrease appears to be the result of muscle wasting, based on reduced gastrocnemius

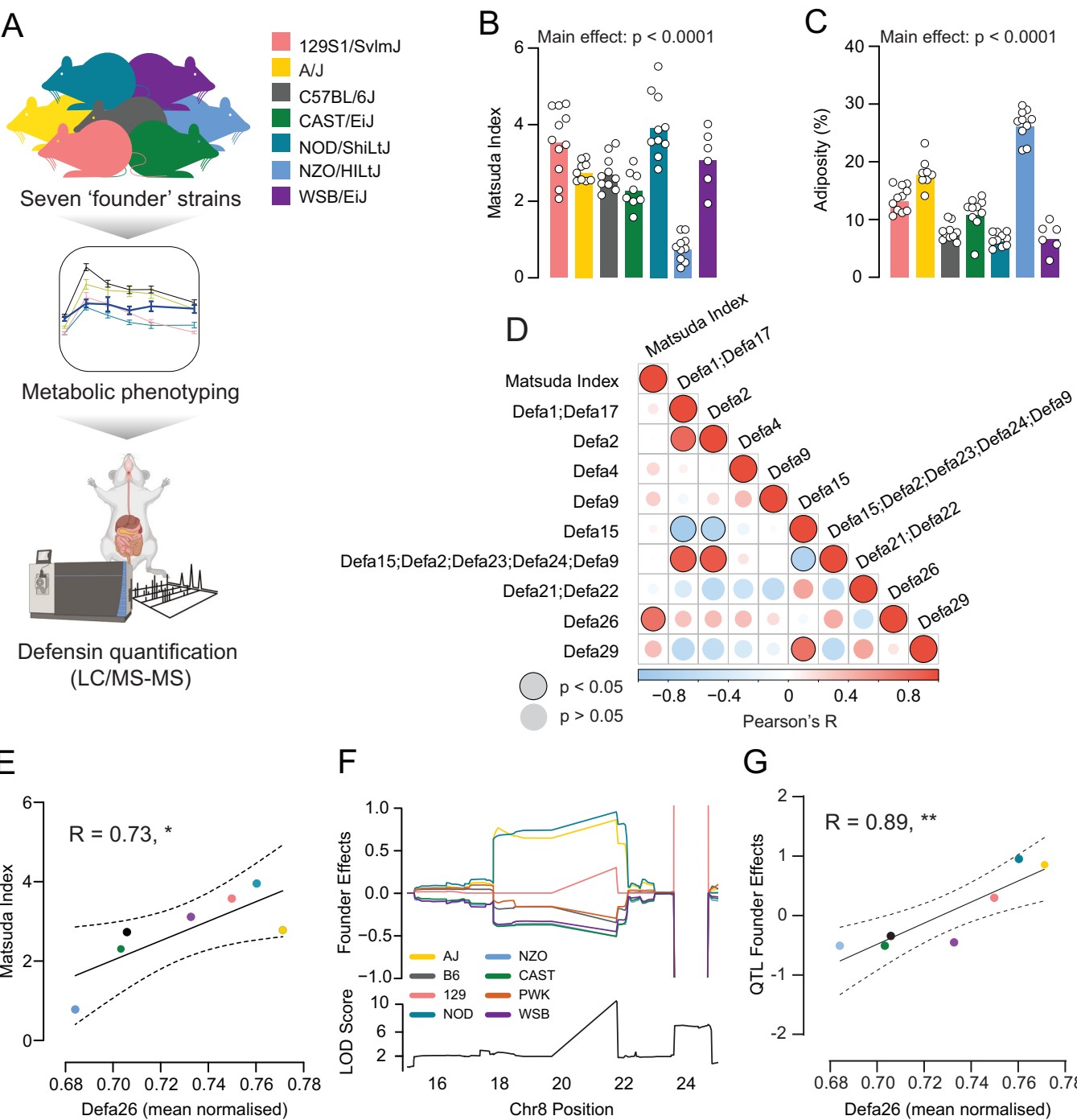

**Figure 3. Analysis of defensin protein expression in Diversity Outbred founder strains.**

(**A**) Schematic of study design to investigate insulin sensitivity and defensin protein expression in small intestines of inbred mouse strains. (**B**) Matsuda Index in Diversity Outbred founder strains. (**C**) Adiposity in Diversity Outbred founder strains. (**D**) Correlations of all quantified defensin peptides (Defa) with Matsuda Index across Diversity Outbred founder strains. (**E**) Correlation of mean normalised alpha-defensin 26 abundances with Matsuda Index in Diversity Outbred founder strains. (**F**) Founder strain contribution estimates for the DOz Matsuda Index QTL (top) and QTL LOD score (bottom) on chromosome eight. (**G**) Correlation of founder strain contribution estimates for the DOz Matsuda Index QTL with mean normalised alpha-defensin 26 abundance. Dashed lines denote 95% confidence intervals. Data are mean with biological replicates shown as individual data points. **$P < 0.01$, *$P < 0.05$ Metabolic phenotypes were compared by one-way ANOVA (**B**, **C**). Pearson's correlations were performed without *P* value adjustment (**D**, **E**, **G**). Source data are available online for this figure.

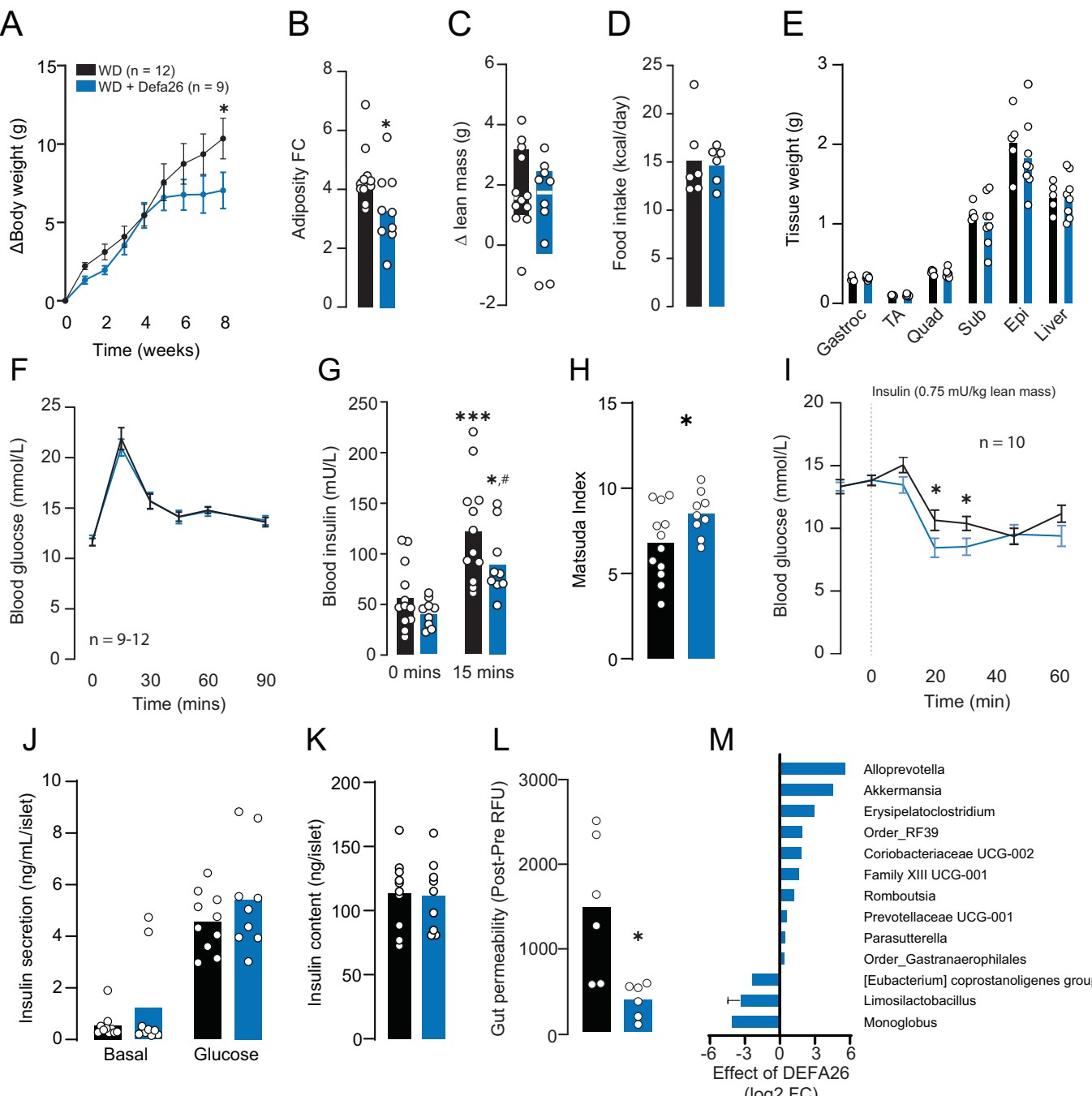

**Figure 4. Metabolic phenotyping of western-diet fed C57BL/6J mice supplemented with alpha-defensin 26.**

(A) Change in body weight (g) of western diet (WD) or WD + alpha-defensin 26 (WD + Defa26) fed C57BL/6J mice over 8 weeks. (B) Relative (fold-change) increase in adipose tissue mass of WD and WD + Defa26 fed C57BL/6J mice over 8 weeks. (C) Change in lean mass (g) of WD and WD + Defa26 fed C57BL/6J mice over 8 weeks. (D) Food intake (kcal/day) of WD and WD + Defa26 fed C57BL/6J mice over 8 weeks. (E) Post-dissection tissue weights of WD and WD + Defa26 fed C57BL/6J mice. (F) Blood glucose and (G) blood insulin concentrations during a glucose tolerance test of C57BL/6J mice fed either a WD or WD + Defa26 for eight weeks. (H) Matsuda Index of C57BL/6J mice fed either a WD or WD + Defa26 for 8 weeks. (I) Blood glucose concentrations during an insulin tolerance test of C57BL/6J mice fed either a WD or WD + Defa26 for 8 weeks. (J) Insulin secretion (K) and content from ex vivo islets collected from C57BL/6J mice fed either a WD or WD + Defa26 for 8 weeks. (L) Relative gut permeability (post – pre FITC fluorescence) of C57BL/6J mice fed either a WD or WD + Defa26 for 8 weeks. (M) Difference (log2 FC) in relative abundance of microbes identified by Analysis of Compositions of Microbiomes with Bias Correction (ANCOM-BC) between C57BL/6J mice fed either a WD or WD + Defa26 for 8 weeks. Data are mean with biological replicates shown as individual data points. For differentially abundant microbes, error bars represent SD of difference between groups. *$P < 0.05$ denotes statistical significance from WD control. Metabolic phenotypes were compared by two-way RM ANOVA with Tukey's LSD (A, F, I), two-way ANOVA with Tukey's LSD (E, G, J), or Student's $t$ test (B–D, H, K, L). Source data are available online for this figure.

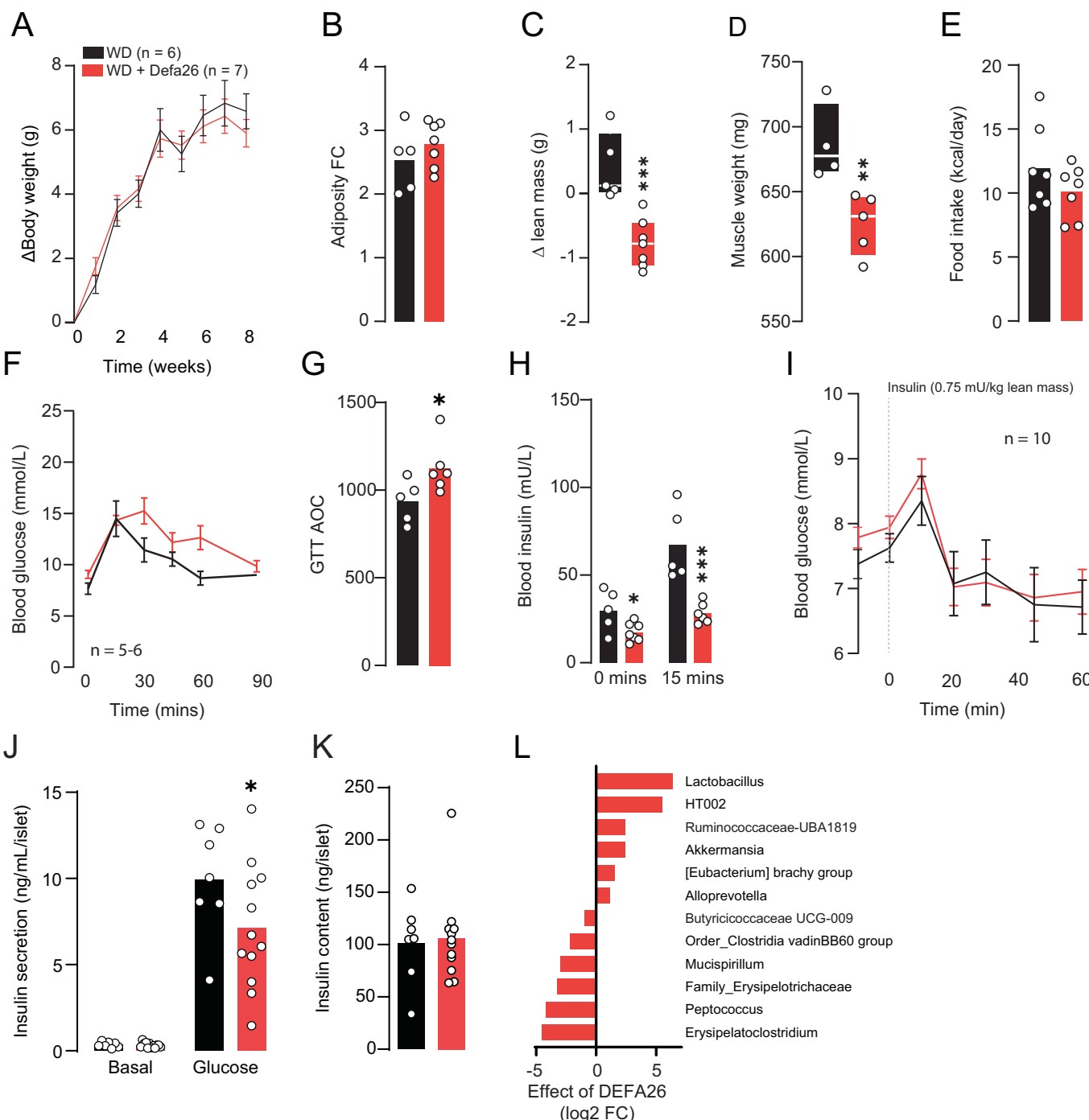

weight (Appendix Fig. S4A), and lowered summed weights of gastrocnemius, tibialis anterior and quadriceps muscles from WD and WD + Defa26 animals (Fig. 5D), notably we observed no difference in C57BL6/J muscle weights (Appendix Fig. S4B). As with C57BL/6J mice, we did not observe a difference in food intake between diets (Fig. 5E).

To assess the effect of Defa26 on glucose homeostasis, we performed GTTs and observed relative fasting hyperglycaemia and glucose intolerance in WD + Defa26 fed A/J mice (Fig. 5F,G). This appeared to be the result of hypoinsulinemia rather than insulin

resistance as WD + Defa26 fed A/J mice exhibited lower circulating insulin levels both in fasting conditions and during a GTT (Fig. 5H), and no difference during an insulin tolerance test (Fig. 5I) relative to WD controls. Consistent with fasting hypoinsulinemia, islets isolated from WD + Defa26 fed A/J mice showed reduced glucose-stimulated insulin secretion compared with WD fed controls (Fig. 5J), despite similar insulin content (Fig. 5K). In contrast to C57BL6/J mice, these findings demonstrates that Defa26 supplementation in A/J mice primarily affects beta-cell secretory function rather than altering peripheral insulin sensitivity.

**Figure 5. Metabolic phenotyping of western-diet fed A/J mice supplemented with alpha-defensin 26.**

(A) Change in body weight (g) of western diet (WD) or WD + alpha-defensin 26 (WD + Defa26) fed A/J mice over 8 weeks. (B) Relative (fold-change) increase in adipose tissue mass of WD and WD + Defa26 fed A/J mice over 8 weeks. (C) Change in lean mass (g) of WD and WD + Defa26 fed A/J mice over 8 weeks. (D) Combined mass of gastrocnemius, tibialis anterior and quadriceps muscles from A/J mice after WD or WD + Defa26 feeding for 8 weeks. (E) Food intake (kcal/day) of WD and WD + Defa26 fed A/J mice. (F) Blood glucose concentrations during a glucose tolerance test of A/J mice fed either a WD or WD + Defa26 for 8 weeks. (G) Glucose tolerance test 'area-under-the-curve' for A/J mice fed either a WD or WD + Defa26 for 8 weeks. (H) Blood insulin concentrations during a glucose tolerance test of A/J mice fed either a WD or WD + Defa26 for 8 weeks. (I) Blood glucose concentrations during an insulin tolerance test of A/J mice fed either a WD or WD + Defa26 for 8 weeks. (J) Insulin secretion from ex vivo islets collected from A/J mice fed either a WD or WD + Defa26 for 8 weeks. (K) Insulin content of islets collected from A/J mice fed either a WD or WD + Defa26 for 8 weeks. (L) Difference (log2 FC) in relative abundance of microbes identified by Analysis of Compositions of Microbiomes with Bias Correction (ANCOM-BC) between A/J mice fed either a WD or WD + Defa26 for 8 weeks. Data are mean with biological replicates shown as individual data points. ***$P < 0.001$, **$P < 0.01$, *$P < 0.05$ denotes statistical significance from WD control. Metabolic phenotypes were compared by two-way RM ANOVA with Tukey's LSD (F, I), two-way ANOVA with Tukey's LSD (A, H, J), or Student's $t$ test (B–E, G, K). Source data are available online for this figure.

Despite these differences relative to C57BL/6J mice *A. muciniphila* and *Alloprevotella spp* were also enriched in the caecums of A/J mice supplemented with Defa26, albeit to a lesser extent (Fig. 5L), although A/J mice fed WD + Defa26 did not exhibit improvements in gut integrity over WD fed controls (Appendix Fig. S4C).

## Disrupted microbial bile acid metabolism may explain the deleterious effects of alpha-defensin 26 supplementation in A/J mice

Despite our hypothesis that Defa26 supplementation would have little to no impact on A/J mice, we in fact observed muscle wasting, glucose intolerance and hypoinsulinemia in response to 8 weeks of dietary supplementation in this strain. This contrasted sharply with the beneficial effects observed in C57BL/6J mice and reinforces the importance of strain selection when testing potential therapeutics. It is possible that strain specific effects of western diet feeding on endogenous Defa26 production/function underpin the differential Defa26 response. For example, if Defa26 is depleted by WD feeding in C57Bl6/J but not A/J mice, dietary supplementation could rescue C57BL6/J mice and potentially harm A/J. To test this, we measured intestinal Defa26 expression in chow and WD fed C57BL6/J and A/J mice. We observed no effect of WD feeding on Defa26 in either strain (Appendix Fig. S5A).

Next, we compared the differentially abundant microbes in C57BL/6J and A/J mice (Fig. 6A,B). Only three microbes were changing concordantly in both strains: *A. muciniphila* (Fig. 6C), *Alloprevotella spp*. (Fig. 6D), and *Erysipelatoclostridium spp* (Fig. 6E). Further reinforcing strain specificity, *Erysipelatoclostridium spp* increased in C57BL/6J but was depleted by Defa26 treatment in A/J mice (Fig. 6E). Interestingly, *Erysipelatoclostridium spp* is the representative taxa for Family_Erysipelatoclostridiaceae, a group enriched in AB mice but not the corresponding Cage mates (Appendix Fig. S1A), suggesting *Erysipelatoclostridium spp* may be a core microbe in defensin-mediated community structure.

To further investigate how microbiome-strain interactions could underpin the differential effects of Defa26 supplementation we compared the microbiomes of Defa26 treated C57BL/6 and A/J mice (Appendix Fig. S5B). This revealed 28 differentially abundant taxa, which were then analysed by taxon-set enrichment analysis (Lu et al, 2023). Using the 'host-intrinsic' dataset revealed a striking enrichment for 'Bile Acid Metabolism', as well as other relevant terms including 'Insulin Resistance' and 'Diabetes Mellitus'

(Fig. 6F). Based on this, and previous work linking microbially derived bile acids to insulin secretion and lean mass (Düfer et al, 2012; Kreznar et al, 2017; Tamai et al, 2022) we set out to profile circulating bile acids in Defa26 fed mice.

Using liquid chromatography-mass spectrometry (LC-MS) we measured the abundance of 16 primary and secondary bile acids in plasma from a subset of C57BL/6J and A/J mice fed either a WD or WD + Defa26 (Appendix Fig. S5C,D). Consistent with strains-specific effects of Defa26 supplementation, principal component analysis separated A/J but not C57BL/6J mice bile acid profiles based on Defa26 supplementation (Fig. 6G). In both C57BL/6J and A/J mice, Defa26 supplementation lowered circulating taurocholic acid (TCA), with a more pronounced effect in C57BL6/J mice (Fig. 6H; Appendix Fig. S5C). Although TCA is a primary bile acid in mice, reductions in circulating levels sampled from the periphery (tail vein) likely reflect microbial rather than hepatic metabolism as only 5% of bile acids escape enterohepatic metabolism. In A/J mice, Defa26 also had striking effect on circulating deoxycholic acid (DCA) and taurodeoxycholic acid (TDCA) (Fig. 6H; Appendix Fig. S5D). Taken together, these changes are indicative of increased deconjugation of TCA in both strains but a reduction in secondary bile acid synthesis in A/J mice. Notably, both DCA and TDCA are derived from microbially mediated deconjugation reactions and are potent agonists for takeda-G-protein-receptor-5 (Tgr5; Gpbar1), a bile acid receptor expressed in skeletal muscle and beta-cells where it regulates hypertrophy and insulin secretion, respectively. Therefore, reductions in DCA and TDCA in A/J mice may explain the observed effects on reduced muscle mass and insulin secretion.

## Discussion

Functional links between the gut microbiome and glucose homeostasis have predominantly been made via high fat/high sugar, low fibre 'western' diets, which bias gut microbial composition towards an inflammatory obesogenic state (Dabke et al, 2019; Liu et al, 2022; Takeuchi et al, 2023b). Here we reveal an alternative gut-microbiome/metabolic health axis by performing genetic mapping in a population of chow-fed DOz mice. We identified a striking insulin sensitivity QTL within the defensin gene cluster, which associated with an enrichment for metabolically beneficial microbes, and correlated with increased expression of the antimicrobial peptide Defa26 in inbred founder strains. We validated this observation by performing dietary supplementation studies in two inbred mouse strains with differential endogenous

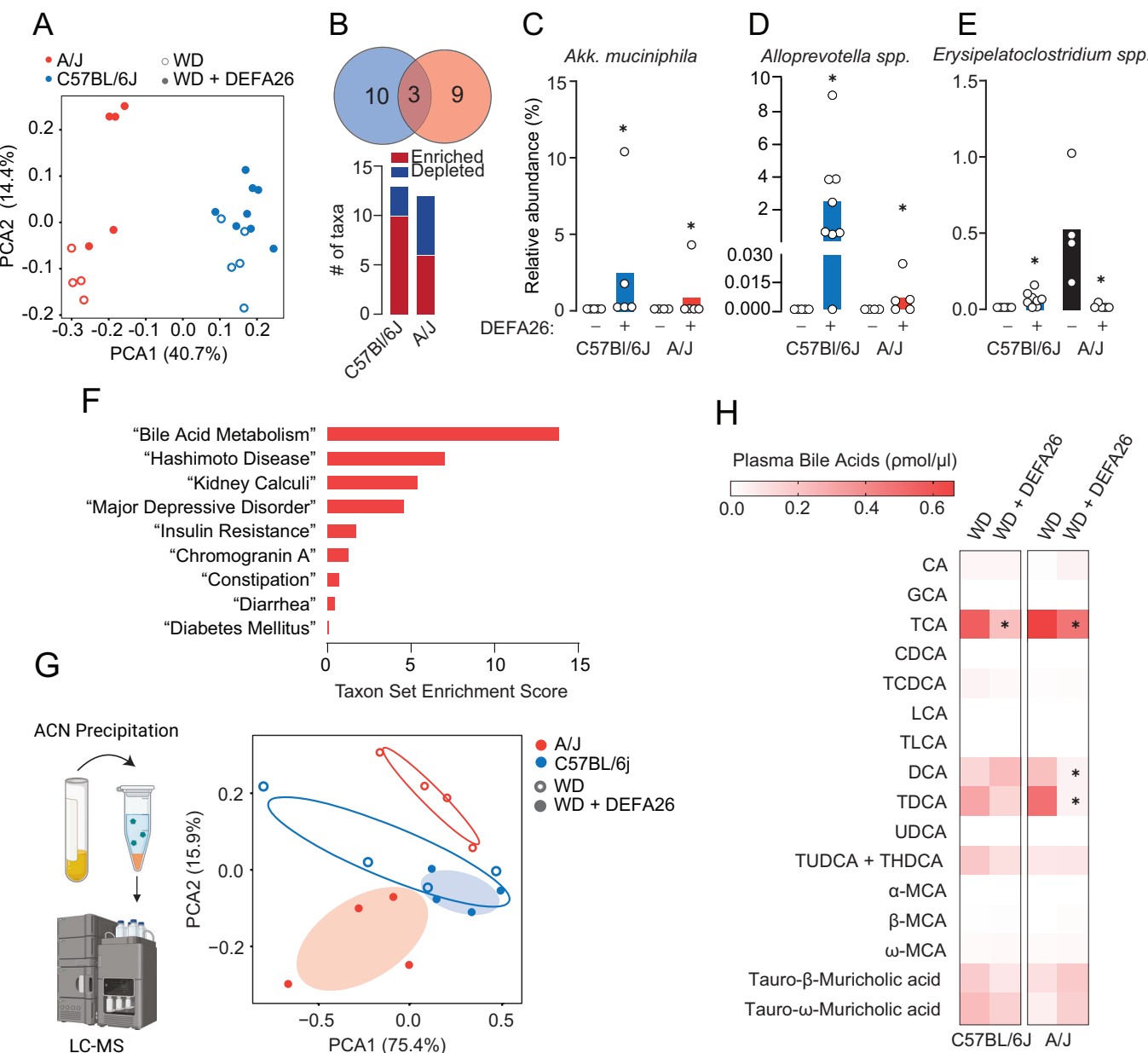

**Figure 6. Comparison of caecal microbiomes and circulating bile acids from C57BL/6J and A/J following alpha-defensin 26 supplementation.**

(A) Visualisation of beta-diversity (ANCOM-BC) in WD and WD + Defa26 fed C57BL/6J and A/J mice. (B) Comparison of differentially abundant microbial taxa in mice fed either a WD or a WD + Defa26 assessed by ANCOM-BC. (C) Relative abundance of Akkermansia muciniphila (D) Alloprevotella sp. and (E) Erysipelatoclostridium spp. in C57BL/6J and A/J mice fed either a WD or a WD + Defa26. (F) Enrichment scores for statistically significant terms included in Taxon Set Enrichment Analysis. (G) Work-flow to quantify circulating bile acids by LC-MS and principal component visualisation of circulating bile acids in C57BL/6J and A/J mice fed either a WD or a WD + Defa26. (H) Heatmap visualisation of circulating bile acid concentrations in C57BL/6J and A/J mice fed either a WD or a WD + Defa26. Data are mean with biological replicates shown as individual data points. *$P < 0.05$ denotes statistical significance from WD control. Difference in microbial relative abundance (B–E) was identified by Analysis of Compositions of Microbiomes with Bias Correction (ANCOM-BC). Bile acids were compared by two-way ANOVA with Tukey's LSD (H). Source data are available online for this figure.

Defa26 expression. Our results revealed that Defa26 controls glucose homeostasis in an inverted U-shaped relationship. At low levels, increasing concentrations of Defa26 improved insulin sensitivity, gut integrity and *A. muciniphila* abundance, whereas excess Defa26 disrupted microbial bile acid metabolism, leading to insulin secretion defects and muscle wasting. This illustrates the

importance of considering genetic variation in the development of metabolic therapeutics and places the microbiome downstream of host genetics in the control of insulin sensitivity.

In our DOz colony, litters are separated at weaning to avoid confounding genetic diversity with cage-effects. We took advantage of this to test whether the defensin locus/insulin sensitivity

association was microbiome mediated. We observed that sharing a cage with an AB mouse conferred a mild beneficial effect on insulin sensitivity and increased *A. muciniphila* abundance. This is consistent with previous defensin supplementation studies (Ehmann et al, 2019; Li et al, 2023) and microbial transfer via coprophagy, and suggests the mechanism linking the defensin locus to insulin sensitivity is microbiome mediated. This has important implications beyond the present study as siblings of different genotypes are commonly co-housed in traditional transgenic mouse experiments. If there is an effect of genotype on gut microbial composition, any subsequent effect on host-physiology may be masked by microbial transfer between cage-mates. For example, a potential Defa26 knock-out mouse model co-housed with wild-type control mice may appear phenotypically normal provided it maintains a healthy microbiome via coprophagy.

The murine defensin cluster on chromosome 8 contains as many as 75 defensin genes, 12 of which were located within 2 Mb of the insulin sensitivity Matsuda Index QTL. To determine which of these were potentially mediating the association between the defensin locus and insulin sensitivity we performed tandem mass spectrometry on small intestine tissue in DOz founder strains. Taking this approach, only Defa26 associated with insulin sensitivity and QTL founder contributions suggesting that genetic variants within the defensin locus promote both Defa26 expression and insulin sensitivity. Defa26 was first identified in 2004 by a phylogenetic search of the *Mus musculus* genome (Patil et al, 2004). Interestingly, the aforementioned study revealed that while the mature defensin peptide sequence varies between isoforms, the signal peptide and pro-peptide are highly conserved. This suggests that selection has optimised the lumenal activity of defensins rather than their pre-secretion processing. Alignment of the mature defensin peptide sequences identified in our study revealed that Defa26 is unique relative to other defensins at the following residues V63G and K72T. Notably, the glycine at position 63 is near the N-terminus of the first beta sheet and may alter interactions between the defensin 'barrel structure' and microbial membranes, leading to selective antimicrobial activities and the observed effects on insulin sensitivity.

Caeca from AB DOz mice, C57BL/6J and A/J mice supplemented with alpha-defensin 26 were enriched for the mucin-dwelling microbe *A. muciniphila*. As defensins are the most concentrated in the mucin layer (Ganz, 2003), it stands to reason *A. muciniphila* has evolved resistance against defensin peptides that can be exploited by defensin administration for beneficial metabolic effects by enabling *A. muciniphila* growth that may have been prevented by other defensin sensitive microbes (Li et al, 2023; Plovier et al, 2017; Yoon et al, 2021). Intriguingly, Zhang et al (Zhang et al, 2023) identified several significant QTL for *A. muciniphila* in high-fat diet (HFD) fed DO mice. While these QTL did not include the defensin locus, they did include the *Atf3* and *Tifa* loci which are involved in Paneth cell differentiation. Differences between our study and that of Zhang et al, likely reflect differences in diet and population genetic architecture, but nevertheless, they both point towards an important role of Paneth cells and defensins in *A. muciniphila* abundance. Increased abundance of *A. muciniphila* may also explain some of the observed changes in circulating bile acids, a recent preprint (preprint: Lucas et al, 2025) identified that *A. muciniphila* can metabolise up to 80% of available TCA. In both A/J and C57BL6/J mice, Defa26 supplementation appeared to decrease circulating TCA levels. While this could be the result of altered liver synthesis as TCA is primary bile acid in mice, it could also be the result of increased deconjugation of TCA by *A. muciniphila*.

In stark contrast with our hypothesis that A/J mice would not respond to Defa26 supplementation they exhibited hypoinsulinemia, muscle wasting and glucose intolerance. Previous experiments in DO founder strains (Kreznar et al, 2017), and TSEA comparing the gut microbiomes of Defa26 fed C57BL/6J and A/J mice indicated this could be due to disrupted bile acid metabolism. Indeed, circulating levels of, DCA and TDCA were reduced by Defa26 supplementation in A/J but not C57BL/6J mice. Both DCA and TDCA are potent agonists of Tgr5, a bile acid receptor expressed in peripheral tissues including skeletal muscle and pancreatic beta-cells. Mice lacking Tgr5 exhibit muscle wasting (Sasaki et al, 2018; Tamai et al, 2022), and islets treated with Tgr5 agonists exhibit increased insulin secretion (Maczewsky et al, 2018). Decreased DCA and TDCA in response to Defa26 may inhibit bile acid signalling in these tissues, resulting in muscle wasting and hypoinsulinemia, respectively. This disruption in bile acid signalling may not have occurred in C7BL6/J mice as they exhibit lower endogenous Defa26 expression and may have experienced a lower effective dose relative to A/J mice. Further work is needed to validate these data as none of the microbial taxa depleted by Defa26 in A/J mice have been causally linked to DCA or TDCA metabolism.

There are two conclusions we draw from the present study. First, the gut microbiome is a downstream effector of genetic variants which regulate insulin sensitivity. In our data, microbial and metabolic differences between mice fed an identical diet can be explained by genetic variance at a single locus. Historically, determining cause-and-effect between microbes and insulin resistance has been challenging as both are altered by diet. However, by anchoring upon genetics, we can infer causality from genotype to phenotype via the proteome and microbiome. Secondly, and perhaps most importantly, the impact of individual biological differences on potential therapeutic outcomes is significant and must be considered as we move into the era of preclinical precision medicine.

## Methods

**Reagents and tools table**

| Reagent/resource | Reference or source | Identifier or catalog number |
|---|---|---|
| **Experimental models** | | |
| Diversity Outbred in Australia (DOz) | Masson et al, 2023a | |
| C57BL/6J | OzGene/ARC | C57BL/6JOzarc |
| A/J | OzGene/ARC | A/JOzarc |
| 129S1/SvlmJ | Australian BioResources | IMSR_JAX:002448 |
| NOD/ShiLtJ | OzGene/ARC | NOD/ShiLtJOzArc |
| CAST/EiJ | Australian BioResources | IMSR_JAX:000928 |
| NZO/HILtJ | Australian BioResources | IMSR_JAX:002105 |
| WSB/EiJ | Australian BioResources | IMSR_JAX:001145 |

| Reagent/resource | Reference or source | Identifier or catalog number |
| --- | --- | --- |
| **Oligonucleotides and other sequence-based reagents** | | |
| 515F-806R primer set | Simpson et al, 2022 | |
| **Chemicals, enzymes and other reagents** | | |
| Human Alpha-defensin 5 | This study | Peptide synthesis methods |
| Mouse Alpha-defensin 26 | This study | Peptide synthesis methods |
| Insulin Mouse Ultra-Sensitive ELISA | Crystal Chem USA | Cat# 90080 |
| HTRF ultra-sensitive insulin assay | (Cisbio, Revvity) | |
| Chow diet | Gordon's Specialty Stock Feeds | |
| AIN-93 vitamin mix | MP Biomedicals | |
| AIN-93 mineral mix | MP Biomedicals | |
| FastDNA Spin Kit for Feces | MP Biomedicals | |
| FITC-dextran | SIGMA-ALDRICH | FD4-1G |
| Fmoc-protected amino acids | Mimotopes and Novabiochem | |
| Agilent InfinityLab Poroshell 120 EC-C18 column | Agilent | |
| Waters Xbridge C18 column | Waters | |
| Waters Sunfire® C18 column | Waters | |
| Waters 2.7 μm CORTECS C18 column | Waters | |
| **Software** | | |
| GraphPad Prism | GraphPad Software | Version 10.4.2 |
| QTL2 | Broman et al, 2019a | |
| *qvalue* | Dabney et al, 2010 | |
| DIA-NN | Aptila | version 1.8.1 |
| *phyloseq* | McMurdie and Holmes, 2013 | |
| *vegan* | McMurdie and Holmes, 2013 | |
| *microbiome* | Leo Lahti, Sudarshan Shetty et al, 2017 | |
| *ANCOM-BC* | Lin and Peddada, 2020 | |
| DADA2 | Callahan et al, 2016 | |
| SCIEX OS | Sciex | version 3.1.6 |
| **Other** | | |
| EchoMRI-900 | EchoMRI Corporation Pte Ltd | |
| Accu-Chek Glucometer | Roche Diabetes Care | |
| Sciex 7600 Zeno TOF mass spectrometer | Sciex | |
| Biotage SYRO I peptide synthesizer | Biotage | |
| Shimadzu 2020 UPLC-MS | Shimadzu | |
| Shimadzu Nexera X2 LC-30AD pump | Shimadzu | |

| Reagent/resource | Reference or source | Identifier or catalog number |
| --- | --- | --- |
| Waters Acquity UPLC BEH300 | Waters | |
| Waters 2535 Quaternary Gradient system | Waters | |
| Waters 2489 UV/Vis Detector | Waters | |
| Waters Fraction Collector III | Waters | |
| Waters Alliance e2695 HPLC system | Waters | |
| Shimadzu Nexera LC-40 UHPLC | Shimadzu | |
| Eppendorf Concentrator Plus | Eppendorf | |

## Mouse breeding and phenotyping

Male 'Diversity Outbred from Oz' (DOz) mice were bred and housed at the Charles Perkins Centre, University of Sydney, NSW, Australia as previously described (Masson et al, 2023b). The DOz mice used in this study were outbred for 27 to 36 generations and comprised a total of 670 male DOz mice across 9 separate cohorts. Genomic DNA was isolated from each mouse and subjected to SNP genotyping (Morgan et al, 2015), followed by genotyping diagnostics and cleaning as described (Broman et al, 2019b). Experiments were performed in accordance with NHMRC guidelines and under approval of The University of Sydney Animal Ethics Committee, approval numbers #1274 and #1988. To delineate genetic from cage-effects, mice were randomised into cages of 3–5 at weaning. All mice were maintained at 23 °C on a 12-h light/dark cycle (0600-1800) and given ad libitum access to a standard laboratory chow diet containing 16% calories from fat, 61% calories from carbohydrates, and 23% calories from protein or an in-house high-fat high-sugar diet (western diet; WD) containing 45% calories from fat, 36% calories from carbohydrate and 19% calories from protein (3.5%g cellulose, 4.5%g bran, 13%g cornstarch, 21%g sucrose, 16.5%g casein, 3.4%g gelatine, 2.6%g safflower oil, 18.6%g lard, 1.2%g AIN-93 vitamin mix (MP Biomedicals), 4.95%g AIN-93 mineral mix (MP Biomedicals), 0.36%g choline and 0.3%g L-cysteine). Fat and lean mass measures were acquired via EchoMRI-900 (EchoMRI Corporation Pte Ltd, Singapore) at 14 weeks of age. Glucose tolerance was determined by oral glucose tolerance test (GTT) at 14-weeks of age by fasting mice for 6-h (0700–1300 h) before oral gavage of 20% glucose solution in water at 2 mg/kg lean mass. Blood glucose concentrations were measured directly by handheld glucometer (Accu-Chek, Roche Diabetes Care, NSW, Australia) from tail blood 0, 15, 30, 45, 60, 90 min after oral gavage of glucose. Blood insulin levels at the 0- and 15-minute time points were measured by mouse insulin ELISA Crystal Chem USA (Elk Grove Village, IL, USA) according to manufacturer instructions. Blood glucose and insulin levels were integrated into a surrogate measure of whole-body insulin sensitivity using the Matsuda Index:

$$Matsuda\ Index = \frac{10,000}{\sqrt{(Glucose_0 \times Insulin_0) \times (Glucose_{GTT\ mean} \times Insulin_{GTT\ mean})}}$$

## Genetic mapping analysis

Genetic mapping of Matsuda Index was performed in R using the QTL2 package (Broman et al, 2019a) following square root transformation of raw values. The GIGA-MUGA single nucleotide polymorphism array was used as genomic inputs for mapping (Morgan et al, 2015), and a covariate and kinship matrix to account for genetic relatedness amongst the DOz animals. Significance thresholds were established by performing 1000 permutations and set at $P < 0.05$.

## Caecal DNA isolation

Genomic DNA was extracted from caecal contents of mice using the FastDNA Spin Kit for Feces (MP Biomedicals) as per the manufacturers protocol. DNA concentration was measured using the Qubit dsDNA BR assay kit (Invitrogen). Mock preparations covering all steps of the procedure were conducted as contamination process controls.

## 16S rRNA gene amplicon sequencing analysis

16S rRNA gene amplicon sequencing was performed on all caecal DNA samples. Barcoded amplicon libraries spanning the V4 hypervariable region of the 16S rRNA gene were prepared (515F-806R primer set- 515 F: GTGYCAGCMGCCGCGGTAA, 806 R: GGACTACNVGGGTWTC-TAAT) and sequenced using the Illumina MiSeq v2 2 ×250 bp platform at the Ramaciotti Centre for Genomics (UNSW, Sydney, Australia). Raw sequence reads were processed using the *DADA2* R package which involves using error profiles to define Amplicon Sequence Variants (ASVs) (Callahan et al, 2016). ASVs were assigned to taxonomy using a pre-trained naïve Bayes classifier trained on the curated 16S rRNA gene SILVA (v138.1) reference database. Any ASV that was present in fewer than 5% of samples or had less than 0.01% of total reads was filtered from the final dataset prior to downstream analysis. Sequencing depth analyses and rarefaction were performed with the *phyloseq* and *vegan* R package (McMurdie and Holmes, 2013).

Analysis and graphical presentation of the resultant ASV data was performed in R using the packages *phyloseq*, *vegan*, *microbiome* and *ggplot2*. Alpha-diversity metrics were calculated using Inverse Simpson's index. Beta-diversity was assessed on centred-log-ratio transformed ASV counts using Bray-Curtis dissimilarity and UniFrac distance and principal coordinate plots generated from the resultant dissimilarity matrix. PERMANOVA (adonis) using the *vegan* R package was used to assess variance in the distance matrices between groups. Differential abundance analysis was performed using the *ANCOM-BC* R package (Lin and Peddada, 2020).

## Intestinal proteomic sample preparation

C57BL/6J, A/J, NOD/ShiLtJ mice were sourced from OzGene/ARC, while 129S1/SvlmJ, CAST/EiJ, NZO/HILtJ and WSB/EiJ mice were sourced from Australian BioResources. Division of the small intestine into thirds was achieved by folding the small intestine into three equivalent lengths and taking a 1 cm section of tissue from the centre of each third. These pieces tissue representing the foregut, midgut and hindgut of each mouse were combined and snap-frozen in liquid nitrogen. Frozen samples were then boiled in 400 uL of SDC buffer (4% sodium deoxycholate, 100 mM Tris-HCl

pH 8.0) by heating at 95 °C for 10 min at 1000 rpm. Samples were then lysed by sonication for 10 min (30 s on, 30 s off, 70% amplitude protocol). Samples were then heated a second time at 95 °C for 10 min at 1000 rpm before being clarified by centrifugation at 18,000 for 10 min at room temperature. Supernatant was taken as lysate and protein concentration was determined by BCA assay, 10 μg of protein was then prepared as previously described (Nelson et al, 2022). Reduction/alkylation (10 mM TCEP, 40 mM CAA) buffer was added to each sample before incubation for 20 min at 60 °C. Once cooled to room temperature, 0.4 μg trypsin and 0.4 μg LysC was added to each sample and incubated overnight (18 h) at 37 °C with gentle agitation. In all, 30 μL water and 50 μL 1% TFA in ethyl acetate was added to stop digestion and dissolve any precipitated SDC. Samples were prepared for mass spectrometry analysis by StageTip clean up using SDB-RPS solid phase extraction material (Rappsilber et al, 2007; Rappsilber et al, 2007). Briefly, 2 layers of SDB-RPS material was packed into 200 μL tips and washed by centrifugation at $1000 \times g$ for 2 min with 50 μL acetonitrile followed by 0.2% TFA in 30% methanol and then 0.2% TFA in water. In total, 50 μL of samples were loaded to StageTips by centrifugation at $1000 \times g$ for 3 min. Stage tips were washed with subsequent spins at $1000 \times g$ for 3 min with 50 μL 1% TFA in ethyl acetate, then 1% TFA in isopropanol, and 0.2% TFA in 5% ACN. Samples were eluted by addition of 60 μL 60% ACN with 5% $NH_4OH_4$. Samples were dried by vacuum centrifugation and reconstituted in 30 μL 0.1% TFA in 2% ACN.

## Mass spectrometry analysis

Peptides prepared as above (2 mg total), were directly injected using a Shimadzu LC-40 UHPLC onto a 5 cm × 2.1 mm C18 column analytical column (Agilent InfinityLab Poroshell 120 EC-C18, 1.9 μm particles) fitted with a 0.5 cm × 2.1 mm C18 guard column (Agilent InfinityLab Poroshell 120 EC-C18, 1.9 μm particles). Peptides were resolved over a gradient from 3% to 36% acetonitrile over 10 min with a flow rate of 0.8 mL min$^{-1}$. Peptide ionization by electrospray occurred at 5.5 kV, with curtain gas 25, gas 2 25 and gas 3 35. A 7600 Zeno TOF mass spectrometer (Sciex) with CID fragmentation used for MS/MS acquisition. Spectra were obtained in a data-independent acquisition using Zeno SWATH with 50 isolation width windows spanning 400–900 Th. Gas-phase fractionation of a pooled mixture of intestine peptides was performed using 100 Th windows per run, to enable spectral library generation covering the 400–900 Th range. Data files were analyzed using the quantitative DIA proteomics search engine, DIA-NN (version 1.8.1) For spectral library generation, the Uniprot mouse Swissprot database downloaded on the 1st of July 2022 was used. Trypsin was set as the protease allowing for 1 missed cleavage and 1 variable modification. Oxidation of methionine were set as a variable modification. Carbamidomethylation of cystine was set as a fixed modification. Remove likely interferences and match between runs were enabled. Neural network classifier was set to double-pass mode. Protein inference was based on genes. Quantification strategy was set to Robust LC (high accuracy). Cross-run normalization was set to RT-dependent. Library profiling was set to full profiling.

## Peptide synthesis

N,N-dimethylformamide (DMF) and dichloromethane ($CH_2Cl_2$) for peptide synthesis were purchased from RCI Labscan and Merck.

Gradient grade acetonitrile (CH$_3$CN) for high-performance liquid chromatography was purchased from Sigma-Aldrich, and ultrapure water (Type 1) was obtained from a Merck Milli-Q EQ 7000 water purification system. Standard Fmoc-protected amino acids (Fmoc-Xaa-OH), coupling reagents and resins were purchased from Mimotopes or Novabiochem. Fmoc-SPPS was performed manually with these reagents and solvents in polypropylene Teflon-fritted syringes purchased from Torviq and through automated synthesis on a SYRO I peptide synthesizer (Biotage). Buffer salts for folding reactions were purchased from Ajax, Sigma-Aldrich, and Thermofisher and used as received. Glutathione reduced and oxidised were purchased from Sigma-Aldrich and Thermofisher, respectively. All other reagents were purchased from AK Scientific or Merck and used as received.

Electrospray mass spectra (ESI-MS) were obtained using a Shimadzu 2020 UPLC-MS with a Nexera X2 LC-30AD pump, Nexera X2 SPD-M30A UV/Vis diode array detector, and a Shimadzu 2020 mass spectrometer using electrospray ionisation (ESI) operating in positive mode. Separations were conducted using a Waters Acquity UPLC BEH300 (1.7 µm, 2.1 × 50 mm C18 column) with a flow rate of 0.6 mL/min. Spectra are recorded from 300 to 2000 Da.

Reverse-phase high performance liquid chromatography (RP-HPLC) was carried out on a Waters 2535 Quaternary Gradient system, fitted with a Waters 2489 UV/Vis Detector (monitoring at 214 and 280 nm) and a Waters Fraction Collector III. Linear peptides were purified by preparative RP-HPLC using an Xbridge C18 column (5 µm, 19 × 150 mm) at a flow rate of 12 mL/min. A mobile phase of Milli-Q water (Solvent A) and HPLC-grade CH$_3$CN (Solvent B) was employed over a linear gradient with 0.1 vol% TFA (trifluoroacetic acid) as an additive. Folded peptides were purified by semi-preparative RP-HPLC using a Xbridge C18 column (5 µm, 10 × 250 mm) at a flow rate of 4 mL/min. A mobile phase of Milli-Q water (Solvent A) and HPLC-grade CH$_3$CN (Solvent B) was employed over a linear gradient with 0.1 vol% TFA as an additive.

Analytical RP-HPLC was performed on a Waters Alliance e2695 HPLC system equipped with a 2998 PDA detector ($\lambda = 210–400$ nm). Separations were performed on a Waters Sunfire® C18 (5 µm, 2.1 × 150 mm) column at 40 °C with a flow rate of 0.5 mL/min. All separations were performed using a mobile phase of 0.1% TFA in water (Solvent A) and 0.1% TFA in CH$_3$CN (Solvent B) using linear gradients.

Peptides were synthesized on a 50 µmol scale. 2-Chlorotrityl chloride (2-CTC) resin was treated with Fmoc-Xaa-OH (1.2 eq) and *i*-Pr$_2$NEt (4.8 eq) in DCM (4 mL). The C-terminal amino acid of each sequence was used for this loading step, ie. Fmoc-Leu-OH was used for the loading of Defa26 and Fmoc-Arg(Pbf)-OH was used for Defa5. Fmoc loading was determined by measuring the piperidine fulvene adduct from Fmoc deprotection. The loading of the resins were: 0.51 mmol/g for Leu on CTC (Defa26) and 0.50 mmol/g for Arg on CTC (DEFA5).

### General procedure A: automated Fmoc-Solid-phase peptide synthesis (SPPS) – SYRO I automatic peptide synthesizer (Biotage)

In total, 50 µmol of the amino acid loaded resin was treated with a solution of piperidine (40 vol%, 0.8 mL) in DMF for 3 min, drained, before repeat treatment with piperidine (20 vol%, 0.8 mL) in DMF for 10 min. The resin was drained and washed with DMF (4 × 1.2 mL) before addition of a solution of Fmoc-Xaa-OH (200 µmol, 4 eq.) and Oxyma (4.4 eq.) in DMF (400 µL), followed by a solution of *N,N'*-diisopropylcarbodiimide (4 eq.) in DMF

(400 µL). The resin was then agitated at 75 °C for 15 min or 50 °C for 30 min as specified [coupling of Fmoc-His(Trt)-OH and Fmoc-Cys(Trt)-OH were reacted at 50 °C for 30 min in all instances]. The resin was then drained via vacuum and one repeat treatment of the coupling conditions was conducted. The resin was then washed with DMF (4 ×1.2 mL) before being treated with a solution of 5 vol % Ac$_2$O and 10 vol% *i*-Pr$_2$NEt in DMF (1.6 mL) and agitated for 5 min to cap unreacted N-terminal amines on the growing peptide. The resin was then drained and washed with DMF (4 × 1.6 mL). Iterative cycles of this process were repeated until complete peptide elongation was achieved after which the resin was washed with DMF (4 × 5 mL) and CH$_2$Cl$_2$ (5 × 5 mL). HCl counterion exchanges were performed by dissolving the folded peptide in 0.1 M HCl and lyophilising on a freeze drier. This HCl treatment and lyophilisation was repeated six times.

### Synthesis of Defa26

The linear Defa26 sequence (Appendix Fig. S6A) was synthesised according to General Procedure A on 2-CTC resin which was loaded with Fmoc-Leu-OH. The peptide was then cleaved from resin by treatment with 85:5:5:5 v/v/v/v TFA/triisopropylsilane/H$_2$O/ethanedithiol for 2 h at rt. The cleaved solution was collected, dried to ~1 mL under N$_2$ flow and the peptide product was precipitated using diethyl ether (2 × 40 mL) and collected *via* centrifugation. The crude linear peptide was then purified by preparative RP-HPLC (0 vol% CH$_3$CN + 0.1 vol% TFA for 10 min, then 0–50 vol% CH$_3$CN + 0.1 vol% TFA over 50 min) and lyophilised affording the linear Defa26 as a white solid (10.14 mg, 4%).

Linear Defa26 (5.5 mg, 1 eq) was first dissolved in 220 µL of rapid dilution buffer containing TRIS (50 mM), NaCl (150 mM), guanidine.HCl (6 M), and tris(2-carboxyethyl)phosphine (2 mM). This solution was then added gradually to a buffer containing NaHCO$_3$ (200 mM), urea (2 M), GSH (1 mM), and GSSG (0.2 mM) in MilliQ water to make up a 1 mg/mL peptide solution. The folding reaction was left for 16 h without stirring. The folding progress was monitored through LC-MS, and a loss of 6 Da and a simultaneous retention time shift indicated completion of folding. The folded peptide (Appendix Fig. S6B) was then purified by semi-preparative RP-HPLC (0 vol% CH$_3$CN + 0.1% TFA for 20 min, then 0–50 vol% CH$_3$CN + 0.1 vol% TFA over 50 min), affording the folded Defa26 as a white solid after lyophilization (1.07 mg, 19% isolated yield). Prior to biological assays, the peptide was converted to the HCl salt through a HCl counterion exchange.

### Synthesis of DEFA5

The linear DEFA5 sequence (Appendix Fig. S6C) was synthesised according to General Procedure A on 2-CTC resin which was loaded with Fmoc-Arg(Pbf)-OH. The peptide was then cleaved from resin by treatment with 85:5:5:5 v/v/v/v TFA/triisopropylsilane/H$_2$O/ethanedithiol for 2 h at rt. The cleaved solution was collected, dried to ~1 mL under N$_2$ flow and the peptide product was precipitated using diethyl ether (2 × 40 mL) and collected via centrifugation. The crude linear peptide was then purified by preparative RP-HPLC (0 vol% CH$_3$CN + 0.1 vol% TFA for 10 min, then 0–50 vol% CH$_3$CN + 0.1 vol% TFA over 50 min), affording the linear DEFA5 as a white solid after lyophilisation (3.19 mg, 3%).

Linear DEFA5 (2.51 mg, 1 eq) was first dissolved in 100 µL of rapid dilution buffer containing TRIS (50 mM), NaCl (150 mM),

guanidine.HCl (6 M), and tris(2-carboxyethyl)phosphine (2 mM). This solution was then added gradually to a buffer containing $NH_4OAc$ (330 mM), guanidine.HCl (500 mM), GSH (1 mM), and GSSG (0.2 mM) in MilliQ water to make up a 1 mg/mL peptide solution. The folding reaction was then left for 40 h without stirring. The folding progress was determined through LC-MS, and a loss of 6 Da and a simultaneous retention time shift indicated folding. The folded peptide (Appendix Fig. S6D) was then purified by semi-preparative RP-HPLC (0 vol% $CH_3CN$ + 0.1% TFA for 20 min, then 0–50 vol% $CH_3CN$ + 0.1 vol% TFA over 50 min), affording the folded DEFA5 as a white solid after lyophilisation (0.62 mg, 24% isolated yield). Prior to biological assays, the peptide was converted to the HCl salt through a HCl counterion exchange.

## Defensin feeding experiments

Prior to allocation into experimental groups, C57BL/6J and A/J mice underwent baseline metabolic phenotyping was described above. Cages of mice from each strain were then randomly assigned to treatment group using a random number generator and fed a WD or a WD containing the luminal forms of either murine alpha-defensin 26 (Defa26) or human alpha-defensin 5 (DEFA5) for 8 weeks. Synthetic peptides were mixed into mouse diet by hand at a final concentration of 10 ng/g based on previous work (Larsen et al, 2019). Even distribution of peptides in food was monitored by the addition of blue food dye which was used as a proxy for the distribution of peptides throughout each batch. After 8 weeks mice underwent a second bout of metabolic phenotyping. Insulin tolerance was determined by fasting mice for 4 h (0900–1300) and administering 0.75 mU/kg lean mass by intraperoteineal injection. Blood glucose concentrations were determined −10, 0, 10, 20, 30, 45, 60 min after insulin injection.

## Islet Isolation

Mouse islets were isolated as per previously described (Yau et al, 2021). In brief, 2 mL of 0.25 mg/mL liberase was injected via the hepatic duct to inflate the pancreas, which was then excised and subject to Histopaque gradient. Islets were handpicked into islet wash buffer (HBSS, 10 mM HEPES, 0.1% BSA) and recovered in Islet Media (RPMI 1640, 10% FBS, 1% penicillin/streptomycin) for 1 h prior to static glucose stimulated secretion assay. Operator was blinded to treatment group during both harvest and picking step.

## Static glucose-stimulated insulin secretion

Recovered islets were placed in 2.8 mM glucose Krebs-Ringer buffer supplemented with 10 mM HEPES (KRBH) in 3.5-cm untreated petri dishes for 30 min (pre-basal). Ten islets per condition, in triplicate, were then handpicked into individual wells of a 96-well plate, containing 150 μL KRBH containing 2.8 mM glucose (basal) for 1 h, then KBRH containing 16.7 mM glucose for 1 h (stimulation). Supernatants were collected, and islets snap-frozen in 50 μL islet lysis buffer (100 nM Tris, 300 mM NaCl, 10 mM NaF, 2 mM sodium orthovanadate). Insulin secretion and total islet insulin content was analysed by commercial HTRF ultra-sensitive insulin assay (Cisbio, Revvity).

Assessment of gut permeability via FITC-dextran oral gavage was conducted as previously described (Tagesson et al, 1978).

Briefly mice were fasted for 4 h (0900–1300) before a baseline blood sample (50 μL) was taken from a tail incision. Mice were then gavaged with 150 μl of 80 mg/ml FITC dextran (4 kDa). After 4 h, a second blood sample was taken. Both samples (baseline and post-gavage were then centrifuged at 5000 rpm for 10 min. Resulting plasma was then diluted 1:10 in PBS and fluorescence was measured at 530 nm with excitation at 485 nm. Data was then expressed as relative fluorescence units. For bile acid quantification, plasma was collected between 1400 and 1600 h.

## Bile acid extraction

In all, 50 μL of plasma thawed on ice was added to 150 μL of ice-cold acetonitrile containing 5 pmoles of d4-cholic acid internal standard. Samples were vortexed for 30 s at maximum speed then centrifuged at 15,000 × g for 10 min at 4 °C to pellet insoluble debris. In total, 170 μL of supernatant was transferred to fused-insert HPLC vials, then vacuum centrifuged to dryness in an Eppendorf Concentrator Plus. Samples were reconstituted in 50 μL of 80:20 water:acetonitrile. All solvents were MS grade.

## Bile acid quantification

Separation of bile acids was performed using a Nexera LC-40 UHPLC (Shimadzu, Rydalmere, NSW, Australia) using a 2.1 × 50 mm, 2.7 μm CORTECS C18 column (Waters, Rydalmere, NSW, Australia) with a 7-min binary gradient of 0.1% formic acid in water (A) and acetonitrile (B) at a flow rate of 0.9 mL/min. Initial gradient conditions of 83:17 A/B rose to 30% B at 1.2 min using curve setting 9. From 1.2 to 3.0 min, the proportion of B increased to 38% using curve −5, then rose to 100% at 4.3 min using curve setting 5. The column was flushed at 100% B for 1.9 min before returning to initial conditions over 0.1 min and being held for 0.7 min. Column temperature was 50 °C and injection volume was 0.5 μL.

MS data were acquired on a ZenoTOF 7600 (Sciex, Mulgrave, VIC, Australia) quadrupole-time-of-flight tandem MS operating with electrospray ionisation in negative polarity. Intact bile acid precursor ions were detected using a TOF MS experiment with mass range 200–600 Da and accumulation time 0.3 s. Source parameters were: Spray voltage: −4500 V, Temperature: 650 °C, Ion source gas 1: 70 psi, Ion source gas 2: 80 psi, Curtain gas: 40 psi. Declustering potential was set to −80 V, collision energy was −10 V and CAD gas was set to 10 (arbitrary units). MS calibration was maintained by Calibrant Delivery System auto-calibration at intervals of ~1 h.

Raw data were acquired in a single batch with acquisition order randomised. Six replicates of a sample pool were distributed through the batch to assess intra-batch imprecision. Six replicate injections from a single vial were acquired to determine instrument repeatability. Data analysis was performed with the Analytics module of SCIEX OS (version 3.1.6). Chromatographic peaks were extracted with a width of 0.02 Da and integrated using the AutoPeak algorithm. Identification of bile acids was based on both accurate precursor *m/z* and retention time matched to commercial standards for all quantified bile acid species (Table EV1). Relative molar amounts for bile acids were calculated by comparing raw peak areas relative to the internal standard. Average mass accuracy was <1 ppm, with a range of +/− 2 ppm across the run. %CVs were calculated as peak area ratios relative to the internal standard.

## Statistical analysis

All analysis and visualisation were performed in either the R programming environment (R Core Team, 2013) or GraphPad Prism (GraphPad Software, San Diego, California USA). For protein correlation analysis the Matsuda Index was calculated using glucose tolerance data before being log2 transformed. To correct for multiple testing, *P* values were adjusted using the *q* value method in the R package *qvalue* (Dabney et al, 2010) unless otherwise stated. Chi-square tests for distribution differences within the data and two/one-way ANOVA tests for group differences were performed in GraphPad Prism.

## Data availability

No large-scale data amenable to database repository deposition were generated in this study. Source data has been included for all main figures. All materials are available upon reasonable request.

The source data of this paper are collected in the following database record: biostudies:S-SCDT-10_1038-S44318-025-00555-5.

## Peer review information

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

## Acknowledgements

We would like to thank the Sydney Mass Spectrometry facility and Large Animal Services in the Charles Perkins Centre at the University of Sydney for mass spectrometry and mouse housing support. This work was supported by the Australian Research Council (Laureate Fellowship) to DEJ and by Diabetes Australia (General Grant) to SWCM.

## Author contributions

**Stewart W C Masson**: Conceptualization; Data curation; Formal analysis; Supervision; Funding acquisition; Validation; Investigation; Visualization; Methodology; Writing—original draft; Project administration; Writing—review and editing. **Rebecca C Simpson**: Resources; Formal analysis; Investigation; Methodology; Writing—review and editing. **Harry B Cutler**: Conceptualization; Data curation; Formal analysis; Writing—review and editing. **Patrick W Carlos**: Resources; Methodology. **Oana C Marian**: Resources; Formal analysis; Methodology. **Belinda Yau**: Resources; Investigation; Methodology. **Meg Potter**: Data curation; Investigation. **Søren Madsen**: Conceptualization; Data curation; Investigation; Writing—review and editing. **Kristen C Cooke**: Supervision; Investigation; Methodology. **Niamh R Craw**: Investigation. **Oliver K Fuller**: Investigation. **Dylan J Harney**: Investigation. **Mark Larance**: Investigation; Methodology. **Gregory J Cooney**: Conceptualization; Investigation; Methodology. **Grant Morahan**: Resources; Writing—review and editing. **Erin R Shanahan**: Supervision; Methodology; Writing—review and editing. **Melkam A Kebede**: Resources; Supervision; Methodology. **Christopher Hodgkins**: Resources; Methodology. **Richard J Payne**: Conceptualization; Resources; Supervision; Methodology. **Jacqueline Stöckli**: Conceptualization; Data curation; Funding acquisition; Investigation; Writing—review and editing. **David E James**: Conceptualization; Resources; Supervision; Funding acquisition; Investigation; Project administration; Writing—review and editing.

Source data underlying figure panels in this paper may have individual authorship assigned. Where available, figure panel/source data authorship is listed in the following database record: biostudies:S-SCDT-10_1038-S44318-025-00555-5.

## Disclosure and competing interests statement

The authors declare no competing interests.

