## [Peer Review File · The EMBO Journal]

Genetic variance in the murine defensin locus modulates glucose homeostasis

Stewart Masson, Rebecca Simpson, Harry Cutler, Patrick Carlos, Oana Marian, Belinda Yau, Meg Potter, Søren Madsen, Kristen Cooke, Niamh Craw, Oliver Fuller, Dylan Harney, Mark Larance, Gregory Cooney, Grant Morahan, Erin Shanahan, Melkam Kebede, Christopher Hodgkins, Richard Payne, Jacqueline Stöckli, and David James

Corresponding authors: Stewart Masson (stewart.masson@sydney.edu.au) , David James (david.james@sydney.edu.au)

Review Timeline:

Submission Date:	6th Nov 24
Editorial Decision:	24th Jan 25
Revision Received:	11th Jun 25
Editorial Decision:	18th Jul 25
Revision Received:	23rd Jul 25
Accepted:	13th Aug 25

Editor: Daniel Klimmeck

Transaction Report:

Dear Dr James, dear Dr Masson,

Thank you again for the submission of your manuscript (EMBOJ-2024-119542) to The EMBO Journal, as well as for your patience with our feedback at this time. As mentioned earlier, your study was assessed by two reviewers with expertise in systemic metabolism and genetics, whose comments are enclosed below.

As you will see from the experts' reports, the referees acknowledge the analysis and potential interest and value of your findings. However, they also express important issues regarding the completeness of your study, which need to be addressed thoroughly to make them supportive of publication in the EMBO Journal. Further, the reviewers raise a number of issues related to the presentation of the findings, additional controls and improved methods annotation required, statistics applied and overall discussion of related literature, that would need to be conclusively addressed to achieve the level of robustness and clarity needed for The EMBO Journal.

Given the overall interest stated and broader angle of your findings, we are able to invite you to revise your manuscript experimentally to address the referees' comments. I need to stress though that we do require strong support from the referees on a revised version of the study in order to move on to publication of the work.

Please feel free to contact me if you have any questions or need further input on the referee comments.

When submitting your revised manuscript, please carefully review the instructions below.

Please feel free to approach me any time should you have additional questions related to this.

Thank you for the opportunity to consider your work for publication.

I look forward to your revision.

Best regards,

Daniel Klimmeck

Daniel Klimmeck, PhD
Senior Editor
The EMBO Journal

Instruction for the preparation of your revised manuscript:

- 1) a .docx formatted version of the manuscript text (including legends for main figures, EV figures and tables). Please make sure that the changes are highlighted to be clearly visible.
- 2) individual production quality figure files as .eps, .tif, .jpg (one file per figure).
- 3) a .docx formatted letter INCLUDING the reviewers' reports and your detailed point-by-point response to their comments. As part of the EMBO Press transparent editorial process, the point-by-point response is part of the Review Process File (RPF), which will be published alongside your paper.
- 4) a complete author checklist, which you can download from our author guidelines ([https://wol-prod-cdn.literatumonline.com/pb-assets/embo-site/Author Checklist%20-%20EMBO%20J-1561436015657.xlsx](https://wol-prod-cdn.literatumonline.com/pb-assets/embo-site/Author%20Checklist%20-%20EMBO%20J-1561436015657.xlsx)). Please insert information in the checklist that is also reflected in the manuscript. The completed author checklist will also be part of the RPF.
- 5) Please note that all corresponding authors are required to supply an ORCID ID for their name upon submission of a revised manuscript.

6) It is mandatory to include a 'Data Availability' section after the Materials and Methods. Before submitting your revision, primary datasets produced in this study need to be deposited in an appropriate public database, and the accession numbers and database listed under 'Data Availability'. Please remember to provide a reviewer password if the datasets are not yet public (see <https://www.embopress.org/page/journal/14602075/authorguide#datadeposition>).

7) Our journal encourages inclusion of *data citations in the reference list* to directly cite datasets that were re-used and obtained from public databases. Data citations in the article text are distinct from normal bibliographical citations and should directly link to the database records from which the data can be accessed. In the main text, data citations are formatted as follows: "Data ref: Smith et al, 2001" or "Data ref: NCBI Sequence Read Archive PRJNA342805, 2017". In the Reference list, data citations must be labeled with "[DATASET]". A data reference must provide the database name, accession number/identifiers and a resolvable link to the landing page from which the data can be accessed at the end of the reference. Further instructions are available at .

8) At EMBO Press we ask authors to provide source data for the main and EV figures. Our source data coordinator will contact you to discuss which figure panels we would need source data for and will also provide you with helpful tips on how to upload and organize the files.

Numerical data can be provided as individual .xls or .csv files (including a tab describing the data). For 'blots' or microscopy, uncropped images should be submitted (using a zip archive or a single pdf per main figure if multiple images need to be supplied for one panel). Additional information on source data and instruction on how to label the files are available at .

9) We replaced Supplementary Information with Expanded View (EV) Figures and Tables that are collapsible/expandable online (see examples in <https://www.embopress.org/doi/10.15252/emj.201695874>). A maximum of 5 EV Figures can be typeset. EV Figures should be cited as 'Figure EV1, Figure EV2' etc. in the text and their respective legends should be included in the main text after the legends of regular figures.

11) For data quantification: please specify the name of the statistical test used to generate error bars and P values, the number (n) of independent experiments (specify technical or biological replicates) underlying each data point and the test used to calculate p-values in each figure legend. The figure legends should contain a basic description of n, P and the test applied. Graphs must include a description of the bars and the error bars (s.d., s.e.m.).

We realize that it is difficult to revise to a specific deadline. In the interest of protecting the conceptual advance provided by the

work, we recommend a revision within 3 months (24th Apr 2025). Please discuss the revision progress ahead of this time with the editor if you require more time to complete the revisions.

Referee #1:

In the present study, Masson et al. survey a large number of diverse outbred mice to identify a genetic locus associating to matsuda index from glucose tolerance tests. This locus encodes many defensin genes, where Defa26 is prioritized by proximity to lead snp. By performing a series of analyses segregating animals according to specific founder (A/J or BL/6) alleles, the authors support a mechanism whereby existence of the B6 allele associates to microbiome composition measures. Finally, this mechanism is validated by administering mice of differing sensitive or resistance background recombinant DEFA26, where changes in glucose metabolism and fat mass during a HFD differed between the strains. The effects of DEF26A are further attributed to impacting microbial composition and BA metabolites. This is an elegant demonstration of how population studies can be used to define testable mechanisms of "personalized responses" to physiologic parameters. Further, the diversity of data and analyses that supports the mechanism of DEFA26 operating in discrete contexts substantiates the robustness of the observations. Below are comments, questions and clarifications, most of which could likely be addressed by changes to the text to clarify interpretations.

- 1) from this reviewer's perspective, The most exciting aspect about the paper are that the authors identified a specific mechanism underlying whole body metabolism using genetic diversity And then leveraged these understandings to come up with a way to personalize responses to this mechanism using the same platform. This type of approach really embodies a solid framework for personalizing actionable treatments and mechanisms, which seems rarely done in literature. I believe that the proposed mechanism of defense and alpha 26 altering bile acid composition and changes in microbiome is a reasonable one But seems somewhat correlative. Testing these independently would not likely being informative (clearly BAs like TCA and altering akkermensia abundance alter metabolism), But likely some of the more discreet mechanisms could be clarified from the data that's already in the study. For example, in figure 6, The abundances of bugs incident in each genotype are plotted and different. But what would be the strain that changes the most between the two responses (ex. LFCB6 response / LFC AJ response). Similarly the PCA graph from abundances (7A) she'll very cleanly that pc1 separates by genotype and PC2 separates by DEF26a response. If you rank-ordered the strains by loadings for PC2 this would also give you A list of strain specific bug responses to the treatment. I suppose these might be how they found examples in C D and E, But since there's a lot of information and diversity of 16S sequencing, it seems worthwhile. To clarify systematically.
- 2) The analyses of segregating almost by a single snip and analyzing accordingly is a reasonable one, But obviously, given the average effect size of a single locus/SNP It seems unlikely that This analysis cleanly separates out glucose metabolic phenotypes. Consistent with this, The segregation analyses seem to parse apart insulin secretion In response to glucose but not necessarily the ability to clear glucose from circulation. In general I would probably tone back some aspects of the text, especially when referring to B6/J mice as only harboring the protective SNP(ex. Line 226).
- 3) related to the point above, The others identify the initial locus by LOD scores which uses a whole region to improve statistical power for associations (instead of just a snp by snp analysis) is there any reason to think that the allele (rs231642830) differs from the others in the locus? It would be surprising given that snps and genes within recombination blocks are always strongly correlated. Consistent with this, It looks like another locus on chromosome 2 maps to matsuda index. If segregating by one of the alleles and this locus showed more towards glucose disposal and less towards insulin response, This could really help to demonstrate the complexity of common genetic variation and areas that the authors are focusing into personalize responses predictions
- 4) The cohousing experiments seem interesting, But it's a little bit difficult to follow. It seems like the take-home message is that possessing the AB allele trumps The innate microbiome composition, specifically for matsuda index. It would be interesting to know from this experiment which bugs still maintained the most significant difference between AB and their AA cage-mates. Especially if these SE aligned with some of the significantly changed Def26a species in strains.
- 5) The analysis of the founder strains shown in figure 3 is interesting And highlights diversity of defending levels. I would suggest a little caution and interpreting thuy Pearson correlations given that there are only seven strains to correlate with. Perhaps an equally informative example is just to rank the strains by defensin levels and show. For example it seems like AJ has the highest expression of DEF26a And B6 much lower. This can be potentially helpful in interpreting The different responses later on.
- 6) related to the point above. Regardless of the mechanism of action, One important question is whether The different responses between B6 and AJ are due to baseline amounts of circulating Def26 or whether this process of producing the soluble protein (or it's mechanism of action) becomes dysregulated in a high-fat diet. At this point could be easily addressed by measuring the levels of the protein In circulation and or intestine In the two strains either before (I think this data is already available in the figure) or after the high-fat diet
- 7) since the mechanism of association and potentially Def26 action significantly changes insulin secretion, It might be worthwhile adding some discussion points on pancreatic function. The authors show striking differences in muscle mass which is fascinating, But likely pancreatic insulin secretion might be playing a role as well. One option is to intersect these locus with other studies using diversity outbred mice studying islet function; however, I'm unsure as to how comparable the populations and

associations might be.

Referee #2:

In this manuscript, Masson and coauthors use a systems genetics approach to uncover novel determinants of glucose homeostasis. They subject 670 Diversity Outbred in Australia (DOz) male mice fed chow diet to oral glucose tolerance test, calculate the Matsuda Index as a proxy for whole-body insulin sensitivity and identify a QTL within the defensin locus on chromosome 8. Using liquid chromatography coupled with tandem mass spectrometry, they pinpoint alpha-defensin 26 (Defa26) as the only isoform positively correlated with insulin sensitivity in the small intestine. The authors then administer Defa26 to C57BL/6J and A/J mice fed western diet and report strain-dependent metabolic effects, likely influenced by the differential abundance of endogenous Defa26. Finally, they link the opposing outcomes to alterations in the gut microbiome and propose decreased DCA and TDCA levels in the A/J mice as a possible mechanism underlying the detrimental effects on the muscle.

This is an interesting systems genetics study that underscores the danger of single-strain overreliance. I have several comments that could help clarify a few outstanding questions.

Major comments:

1. The authors conclude insulin sensitivity based on data collected from an oral glucose tolerance test. Did the authors perform insulin tolerance tests or studied the pancreas to rule out strain-dependent differences in insulin production/secretion? Any data addressing these points, even on the smaller study with the C57BL/6J and A/J mice, would strengthen the study.
2. The cohousing experiments presented in Figure 2 are unclear, with sometimes opposite effects on the microbiome and no effects on insulin sensitivity. Can the authors comment on this?
3. For the Pearson correlation analysis, have the authors checked the adjusted P value? Is the Defa26 and Matsuda index still significantly correlated after multiple-test adjustment?
4. What is the rationale for using A/J mice in the validation experiment. Based on the QTL effect panel (Figure 3F), the DOz mice that carried the A/J allele have higher insulin sensitivity, but A/J mice have lower insulin sensitivity.
5. There is a trend in lean mass decrease in C57BL/6J mice (Figure 4C). Can the authors include muscle weight data, as they did for the A/J mice?
6. The bile acid data is interesting but premature, and this should be discussed in the text. While the authors link microbiome changes to differences in bile acid levels, TCA - a primary bile acid - is decreased in both mouse models. Furthermore, there is no evidence that bacteria able to convert primary into secondary bile acids are affected by Defa26 supplementation. In addition, DCA and TDCA are more potent agonists for TGR5 than for FXR and this should at least be discussed with appropriate references.
7. The data resource table or the data available statement should be provided.
8. The statistical tests used for each panel should be reported in every figure legend. Is there nothing significant in Figures 3B, 3C, 4A, 4I and 5J?

Minor comments:

1. There are 670 mice in total, but the AA genotype mice is 657 and AB genotype mice is 14, then it is 671 mice in total.
2. Is Figure S2A referring to Defa5 or Defa26 as indicated in the figure?
3. For the Figure 6D, is this bacterium also significantly changed in A/J?
4. In line 431, AB DOz mice were not supplemented with alpha-defensin 26 in this study and the level of alpha-defensin 26 in the AB DOz was also not shown.
5. Essential details in the Methods sections are missing and should be added, for instance, reference of the diets used, concentration of the peptides admixed to the diet, feeding status, and time of collection of the plasma for the bile acid analysis.

Referee #1:

In the present study, Masson et al. survey a large number of diverse outbred mice to identify a genetic locus associating to matsuda index from glucose tolerance tests. This locus encodes many defensin genes, where Defa26 is prioritized by proximity to lead snp. By performing a series of analyses segregating animals according to specific founder (A/J or BL/6) alleles, the authors support a mechanism whereby existence of the B6 allele associates to microbiome composition measures. Finally, this mechanism is validated by administering mice of differing sensitive or resistance background recombinant DEFA26, where changes in glucose metabolism and fat mass during a HFD differed between the strains. The effects of DEF26A are further attributed to impacting microbial composition and BA metabolites. This is an elegant demonstration of how population studies can be used to define testable mechanisms of "personalized responses" to physiologic parameters. Further, the diversity of data and analyses that supports the mechanism of DEFA26 operating in discrete contexts substantiates the robustness of the observations. Below are comments, questions and clarifications, most of which could likely be addressed by changes to the text to clarify interpretations.

1) from this reviewer's perspective, The most exciting aspect about the paper are that the authors identified a specific mechanism underlying whole body metabolism using genetic diversity And then leveraged these understandings to come up with a way to personalize responses to this mechanism using the same platform. This type of approach really embodies a solid framework for personalizing actionable treatments and mechanisms, which seems rarely done in literature. I believe that the proposed mechanism of defense and alpha 26 altering bile acid composition and changes in microbiome is a reasonable one But seems somewhat correlative. Testing these independently would not likely being informative (clearly BAs like TCA and altering akkermensia abundance alter metabolism), But likely some of the more discreet mechanisms could be clarified from the data that's already in the study. For example, in figure 6, The abundances of bugs incident in each genotype are plotted and different. But what would be the strain that changes the most between the two responses (ex. LFCB6 response / LFC AJ response). Similarly the PCA graph from abundances (7A) she'll very cleanly that pc1 separates by genotype and PC2 separates by DEF26a response. If you rank-ordered the strains by loadings for PC2 this would also give you A list of strain specific bug responses to the treatment. I suppose these might be how they found examples in C D and E, But since there's a lot of information and diversity of 16S sequencing, it seems worthwhile. To clarify systematically.

With respect to mouse strain specific microbiome responses to Defa26, this information can be found in figure 6F. Blue bars represent microbes which are enriched in Defa26 treated B6 mice relative to Defa26 treated A/J mice, vice versa for the red bars

(enriched in Defa26 treated A/J mice relative to Defa26 treated B6 mice). The figure, and figure legend have been updated to make this clearer.

The figure presented in 6A is a Principal Coordinates Analysis (PCoA) rather than Principal Component Analysis (PCA). This is the method of choice for presenting beta diversity between microbiome samples calculated by Bray-Curtis dissimilarity. Rather than calculated on raw ASV abundances, the PCoA presented is an ordination of distance values and cannot be used to visualise loadings as you would in a standard PCA plot. However, the information the reviewer is requesting (strain specific responses to Defa26) can be found in Supplementary Figure 5B. These were identified by systematic pairwise analysis of compositions of microbiomes with bias correction (ANCOM-BC), a method for identifying differences between samples, while accounting for sampling bias (see PMID: 32665548 for more).

The microbes presented in 6C-E were identified by comparing within strain ANCOM-BC results from Figure 4L and 5L and taking the overlap. This approach is visually represented in Figure 6B.

2) The analyses of segregating almost by a single snp and analysing accordingly is a reasonable one, But obviously, given the average effect size of a single locus/SNP It seems unlikely that This analysis cleanly separates out glucose metabolic phenotypes. Consistent with this, The segregation analyses seem to parse apart insulin secretion In response to glucose but not necessarily the ability to clear glucose from circulation. In general I would probably tone back some aspects of the text, especially when referring to B6/J mice as only harboring the protective SNP(ex. Line 226).

During revision we noted that the SNP used was incorrectly labelled and should be listed as rs23754102. This has been updated through the text. Indeed, many SNPs are likely contributing to the observed association, and the use of a single SNP is simply a tool to segregate mice for analysis. We have softened the language surrounding the protective SNP as follows:

Lines 125-128 now read: “We conducted single nucleotide polymorphism (SNP) analysis of the defensin locus and identified several subthreshold SNPs, as well as one SNP (rs23754102) and one structural variant (SV_8_21749161_21749163) with genome-wide significant logarithm of the odds (LOD) scores (Figure 2A).”

Lines 133 to 135 now read: “While we cannot make claims about the causal SNP, these data suggest genetic variance at the defensin locus is linked to improved whole body insulin sensitivity.”

3) related to the point above, The others identify the initial locus by LOD scores which uses a whole region to improve statistical power for associations (instead of just a snp by snp analysis) is there any reason to think that the allele (rs231642830) differs from

the others in the locus? It would be surprising given that snps and genes within recombination blocks are always strongly correlated. Consistent with this, It looks like another locus on chromosome 2 maps to matsuda index. If segregating by one of the alleles and this locus showed more towards glucose disposal and less towards insulin response, This could really help to demonstrate the complexity of common genetic variation and areas that the authors are focusing into personalize responses predictions

The SNP probabilities for the chromosome 2 QTL were too low to confidently segregate mice for further analysis. However, this QTL is within the *Ptprt* (Receptor-type tyrosine-protein phosphatase T) gene. Interestingly, *Ptprt* has been shown to regulate insulin sensitivity (PMID: 24949727). Whole body knock-out of *Ptprt* reduces food intake and is protective against high-fat diet feeding induced insulin resistance and glucose intolerance. We have added this to the manuscript and restructured figures 1 and 2, including labelling *Ptprt* in Figure 1B.

Lines 107-119 now reads: “The QTL on chromosome 2 centred within *Ptprt*, the gene encoding receptor-type tyrosine-protein phosphatase T. Interestingly, whole body knockout of *Ptprt* is protective against high-fat diet induced insulin resistance (42), and receptor-type tyrosine-protein phosphatase T has been shown to regulate the stability of catenin proteins (43), several of which have been linked to glucose homeostasis (44-49). The chromosome 8 locus centred over the defensin gene cluster, a syntenic region shared between mice and humans (50), which contains 53 defensin genes and 22 defensin pseudogenes (Figure 1B). In mice, defensins are secreted from Paneth cells in the intestinal crypt into the gut lumen to modulate microbial composition (Figure 1C). Based on previous work (26-29), the link between the microbiome and metabolic health, and on-going interest in gut-derived peptides as therapeutics for metabolic disease, we selected this QTL for further validation.”

4) The cohousing experiments seem interesting, But it's a little bit difficult to follow. It seems like the take-home message is that possessing the AB allele trumps The innate microbiome composition, specifically for matsuda index. It would be interesting to know from this experiment which bugs still maintained the most significant difference between AB and their AA cage-mates. Especially if these SE aligned with some of the significantly changed Def26a species in strains.

We have tried to improve clarity of this section.

Lines 137-146 now read: “Based on defensin’s role in modulating the microbiome (22, 24), we performed 16S rRNA sequencing on the caecal microbiomes of three groups: 1) mice carrying the putative protective rs23754102 allele (AB), 2) cage mates of these mice, and 3) a subset of AA control mice that neither carry the protective rs23754102 allele, nor share a cage with a mouse that does (Figure 2B). Because mice are coprophagic and cage mates are distantly related, not littermates as are common in

inbred strains, we can compare insulin sensitivity and microbial composition between cage mates and AA control mice. Evidence of increased insulin sensitivity or increased abundance of metabolically beneficial microbes in cage mates of AB would support a microbiome mediated mechanism of action.”

Eleven taxa were differentially abundant between AB and their cage mates, three of these *Blautia*, *Oscillospira* and *Family_Erysipelatoclostridiaceae* exhibited patterns between AA, cage mate and AB mice consistent with a microbial transfer from AB to their cage mates. What is most exciting is that *Family_Erysipelatoclostridiaceae* was also among the most changed microbes upon Defa26 feeding in B6 and AJ mice (Fig 6E). We have included this analysis in the manuscript as a new supplemental figure (Fig S1) and discussed in the text.

Lines 164-174 now read: “Eleven taxa were differentially abundant between AB mice and their cage mates (Supplementary Figure 1A), excluding those which are not different between AA and AB mice, leaving 4 microbes with abundance patterns consistent with microbial transfer (Supplementary Figure 1B-E). *Blautia*, *Oscillospira*, and *Lachnospiraceae-45410* were lower in Cage mates than AA mice, and lower again in AB mice. While *Family_Erysipelatoclostridiaceae* was enriched in AB mice relative to both AA and Cage mates. Out of all differentially abundant microbes, *A. muciniphila* stood out with strong links to metabolic health and has been shown to increase in response to α -defensin administration in C57BL/6J mice (11, 12, 29, 51, 52). These data are consistent with AB mice possessing altered microbiomes relative to AA mice, and this can be transmitted to their cage mates.”

Lines 312-316 now read: “Interestingly, *Erysipelatoclostridium* spp is the representative taxa for *Family_Erysipelatoclostridiaceae*, a group enriched in AB mice but not the corresponding Cage mates (Supplementary Figure S1E), suggesting *Erysipelatoclostridium* spp may be a core microbe in defensin-mediated community structure.”

5) The analysis of the founder strains shown in figure 3 is interesting And highlights diversity of defending levels. I would suggest a little caution and interpreting thuy Pearson correlations given that there are only seven strains to correlate with. Perhaps an equally informative example is just to rank the strains by defensin levels and show. For example it seems like AJ has the highest expression of DEF26a And B6 much lower. This can be potentially helpful in interpreting The different responses later on.

Comparison of defensin levels between strains can be made as the reviewer has done with figure 3. We appreciate that performing correlations with only 7 mouse strains has limited power, but given we subsequently performed validation experiments using Defa26 we have confidence in our findings. However, to highlight this limitation we have amended the text.

Lines 199-202 now read: “This analysis is likely underpowered. Previous work has highlighted relationships between alternative defensin isoforms and metabolic health (28, 29, 53), so with a larger panel of mouse strains it would be possible to identify other defensin isoforms which associate with insulin sensitivity.”

6) related to the point above. Regardless of the mechanism of action, One important question is whether The different responses between B6 and AJ are due to baseline amounts of circulating Def26 or whether this process of producing the soluble protein (or it's mechanism of action) becomes dysregulated in a high-fat diet. At this point could be easily addressed by measuring the levels of the protein In circulation and or intestine In the two strains either before (I think this data is already available in the figure) or after the high-fat diet

Unlike humans, mouse defensin peptides are not expressed by circulating immune cells and therefore are not found in the circulation. At the reviewer’s suggestion we quantified intestinal alpha-defensin 26 expression in B6 and A/J mice after high-sugar/high-fat western feeding and observed comparable levels to chow fed mice. This suggests that strain-specific diet-induced changes in Defa26 expression do explain the strain specific response to Defa26 supplementation. We have added these data to the manuscript as Figure S4A and discussed these as part of the comparison between B6 and A/J response to Defa26.

Lines 299-305 now read: “It is possible that strain specific effects of western diet feeding on endogenous Defa26 production/function underpin the differential Defa26 response. For example, if Defa26 is depleted by WD feeding in C57BL6/J but not A/J mice, dietary supplementation could rescue C57BL6/J mice and potentially harm A/J. To test this, we measured intestinal Defa26 expression in chow and WD fed C57BL6/J and A/J mice. We observed no effect of WD feeding on Defa26 in either strain (Supplementary Figure 5A).”

7) since the mechanism of association and potentially Def26 action significantly changes insulin secretion, It might be worthwhile adding some discussion points on pancreatic function. The authors show striking differences in muscle mass which is fascinating, But likely pancreatic insulin secretion might be playing a role as well. One option is to intersect these locus with other studies using diversity outbred mice studying islet function; however, I'm unsure as to how comparable the populations and associations might be.

We attempted to integrate our genetic mapping data with previous work by the Attie Lab on insulin secretion genetic architecture and with the Auwerx’s Lab’s work on bile acids using <https://qtlviewer.jax.org/>. However, neither produced any overlapping SNPs with our own. This may be the result of differences in diet as, in contrast to their mice which were WD fed, our mice are chow-fed.

However, we have addressed pancreatic insulin secretion directly by performing ex vivo glucose stimulated insulin secretion assays in C57BL6/J and A/J mice fed either a WD or a WD + Defa26. These data can be found in Figure 4J,K and Figure 5J,K. Consistent with in vivo insulin data, A/J but not C57BL6/J mice exhibited attenuated insulin secretion when fed a WD supplemented with Defa26 for 8 weeks. Neither strain exhibited differences in total islet insulin content suggesting the defect is restricted to secretion rather than production. These data are also consistent with previous reports linking bile acids to insulin secretion, potentially via Tgr5 signalling.

Lines 233-239 now read: “To test this further we conducted insulin tolerance tests and measured insulin secretion in ex vivo islets in WD, and WD + Defa26 fed C57BL6/J mice (Figure 4I). We observed a greater suppression of glucose 20-30 minutes after intraperitoneal insulin injection in mice supplemented with defensin relative to WD fed controls but no difference in basal or glucose-stimulated insulin secretion, or total islet insulin content (Figure 4J,K). This strongly suggests Defa26 supplementation increases insulin sensitivity in C57BL6/J mice.”

Lines 278-286 now read: “This appeared to be the result of hypoinsulinemia rather than insulin resistance as WD + Defa26 fed A/J mice exhibited lower circulating insulin levels both in fasting conditions and during a GTT (Figure 5H), and no difference during an insulin tolerance test (Figure 5I) relative to WD controls. Consistent with fasting hypoinsulinemia, islets isolated from WD + Defa26 fed A/J mice showed reduced glucose-stimulated insulin secretion compared with WD fed controls (Figure 5J), despite similar insulin content (Figure 5K). In contrast to C57BL6/J mice, these findings demonstrates that Defa26 supplementation in A/J mice primarily affects beta-cell secretory function rather than altering peripheral insulin sensitivity.”

Referee #2:

In this manuscript, Masson and coauthors use a systems genetics approach to uncover novel determinants of glucose homeostasis. They subject 670 Diversity Outbred in Australia (DOz) male mice fed chow diet to oral glucose tolerance test, calculate the Matsuda Index as a proxy for whole-body insulin sensitivity and identify a QTL within the defensin locus on chromosome 8. Using liquid chromatography coupled with tandem mass spectrometry, they pinpoint alpha-defensin 26 (Defa26) as the only isoform positively correlated with insulin sensitivity in the small intestine. The authors then administer Defa26 to C57BL/6J and A/J mice fed western diet and report strain-dependent metabolic effects, likely influenced by the differential abundance of endogenous Defa26. Finally, they link the opposing outcomes to alterations in the gut microbiome and propose decreased DCA and TDCA levels in the A/J mice as a possible mechanism underlying the detrimental effects on the muscle.

This is an interesting systems genetics study that underscores the danger of single-strain overreliance. I have several comments that could help clarify a few outstanding questions.

Major comments:

1. The authors conclude insulin sensitivity based on data collected from an oral glucose tolerance test. Did the authors perform insulin tolerance tests or studied the pancreas to rule out strain-dependent differences in insulin production/secretion? Any data addressing these points, even on the smaller study with the C57BL/6J and A/J mice, would strengthen the study.

As per the reviewer's suggestion we have performed both insulin tolerance tests and ex vivo insulin secretion assays in WD fed C57BL/6J and AJ mice with and without Defa26 supplementation. Consistent with our previous data, Defa26 supplemented C57BL/6J mice responded more favourably to insulin than WD fed C57BL/6J mice. In addition, we observed no effect of defensin supplementation on insulin tolerance in A/J mice. Ex vivo insulin secretion was also consistent with our previous data, Defa26 supplementation had no effect on basal or glucose-stimulated insulin secretion in C57BL/6J mice, but significantly attenuated glucose-stimulated insulin secretion in A/J mice. Neither strain exhibited differences in insulin content suggesting a secretion, but not production defect. These data are shown in Figure 4 and Figure 5.

Lines 233-239 now read: "To test this further we conducted insulin tolerance tests and measured insulin secretion in ex vivo islets in WD, and WD + Defa26 fed C57BL6/J mice (Figure 4I). We observed a greater suppression of glucose at 20-30 minutes after intraperitoneal insulin injection in mice supplemented with defensin relative to WD fed controls but no difference in basal or glucose-stimulated insulin secretion, or total islet insulin content (Figure 4J,K). This strongly suggests Defa26 supplementation increases insulin sensitivity in C57BL6/J mice."

Lines 278-286 now read: "This appeared to be the result of hypoinsulinemia rather than insulin resistance as WD + Defa26 fed A/J mice exhibited lower circulating insulin levels both in fasting conditions and during a GTT (Figure 5H), and no difference during an insulin tolerance test (Figure 5I) relative to WD controls. Consistent with fasting hypoinsulinemia, islets isolated from WD + Defa26 fed A/J mice showed reduced glucose-stimulated insulin secretion compared with WD fed controls (Figure 5J), despite similar insulin content (Figure 5K). In contrast to C57BL6/J mice, these findings demonstrates that Defa26 supplementation in A/J mice primarily affects beta-cell secretory function rather than altering peripheral insulin sensitivity."

2. The cohousing experiments presented in Figure 2 are unclear, with sometimes opposite effects on the microbiome and no effects on insulin sensitivity. Can the authors comment on this?

We would not expect identical insulin sensitivity or microbiome composition between AB and cage mates as cage mates are only getting second-hand effects of defensin. Likely a lower dose than AB mice. However, cage-mates of AB mice are still significantly more insulin sensitive than non-cage mate AA control mice. In addition, not every microbial change may be transferred between AB mice and their cage mates. Microbiome community structure is complex, and the presence of dietary and immune factors which increase the relative abundance of one microbe may decrease the abundance of another. Microbes interact with one another, and increasing the abundance of one microbe, for example by increasing defensin expression or consuming faeces enriched with defensins/defensin responsive microbes, may indirectly decrease a second microbe either directly via microbe-microbe interactions or indirectly via resource competition. We have tried to improve clarity of this section and added additional analysis and discussion based on similar comments from reviewer 1.

Lines 137-146 now read: “Based on defensin’s role in modulating the microbiome (22, 24), we performed 16S rRNA sequencing on the caecal microbiomes of three groups: 1) mice carrying the putative protective rs23754102 allele (AB), 2) cage mates of these mice, and 3) a subset of AA control mice that neither carry the protective rs23754102 allele, nor share a cage with a mouse that does (Figure 2B). Because mice are coprophagic and cage mates are distantly related, not littermates as are common in inbred strains, we can compare insulin sensitivity and microbial composition between cage mates and AA control mice. Evidence of increased insulin sensitivity or increased abundance of metabolically beneficial microbes in cage mates of AB would support a microbiome mediated mechanism of action.”

Eleven taxa were differentially abundant between AB and their cage mates, three of these *Blautia*, *Oscillospira* and *Family_Erysipelatoclostridiaceae* exhibited patterns between AA, cage mate and AB mice consistent with a microbial transfer from AB to their cage mates. What is most exciting is that *Family_Erysipelatoclostridiaceae* was also among the most changed microbes upon Defa26 feeding in B6 and AJ mice (Fig 6E). We have included this analysis in the manuscript as a new supplemental figure (Fig S1) and discussed in the text.

Lines 164-174 now read: “Eleven taxa were differentially abundant between AB mice and their cage mates (Supplementary Figure 1A), excluding those which are not different between AA and AB mice leaves 4 microbes with abundance patterns consistent with microbial transfer (Supplementary Figure 1B-E). *Blautia*, *Oscillospira*, and *Lachnospiraceae-45410* were lower in Cage mates than AA mice, and lower again in

AB mice. While Family_Erysipelatoclostridiaceae was enriched in AB mice relative to both AA and Cage mates. Out of all differentially abundant microbes, *A. muciniphila* stood out with strong links to metabolic health and has been shown to increase in response to α -defensin administration in C57BL/6J mice (11, 12, 29, 51, 52). These data are consistent with AB mice possessing altered microbiomes relative to AA mice, and this can be transmitted to their cage mates.”

Lines 312-316 now read: “Interestingly, *Erysipelatoclostridium* spp is the representative taxa for Family_Erysipelatoclostridiaceae, a group enriched in AB mice but not the corresponding Cage mates (Supplementary Figure 1A), suggesting *Erysipelatoclostridium* spp may be a core microbe in defensin-mediated community structure.”

3. For the Pearson correlation analysis, have the authors checked the adjusted P value? Is the Defa26 and Matsuda index still significantly correlated after multiple-test adjustment?

We have not adjusted p-values for multiple testing of defensin peptide expression vs Matsuda Index. This has been added to the figure legend for transparency.

4. What is the rationale for using A/J mice in the validation experiment. Based on the QTL effect panel (Figure 3F), the DOz mice that carried the A/J allele have higher insulin sensitivity, but A/J mice have lower insulin sensitivity.

In our data, chow-fed A/J and C57BL/6J mice have equivalent insulin sensitivity (Figure 3B). Our initial hypothesis was that A/J mice would serve as negative control as they have higher levels of Defa26 expression and are well known to be protected from western diet induced insulin resistance.

This rationale is found on lines 261-264: “In view of the responses observed in C57BL/6J mice, we next performed experiments in A/J mice. A/J mice are protected from diet-induced insulin resistance (30, 55) and express relatively high levels of Defa26 (Figure 3E), and so we hypothesised that dietary supplementation in this strain would have no effect on whole-body metabolism.”

5. There is a trend in lean mass decrease in C57BL/6J mice (Figure 4C). Can the authors include muscle weight data, as they did for the A/J mice?

We have included tissue weight, including summed muscle weights for C57BL/6J mice in Figure 4E and Supplementary Figure 4.

Line 229-230 now reads: “We also observed no difference in specific tissue weights (Figure 4E).”

Lines 268-273 now reads: “This decrease appears to be the result of muscle wasting, based on reduced gastrocnemius weight (Supplementary Figure 4A), and lowered

summed weights of gastrocnemius, tibialis anterior and quadriceps muscles from WD and WD + Defa26 animals (Figure 5D), notably we observed no difference in C57BL6/J muscle weights (Supplementary Figure 4A).”

6. The bile acid data is interesting but premature, and this should be discussed in the text. While the authors link microbiome changes to differences in bile acid levels, TCA - a primary bile acid - is decreased in both mouse models. Furthermore, there is no evidence that bacteria able to convert primary into secondary bile acids are affected by Defa26 supplementation. In addition, DCA and TDCA are more potent agonists for TGR5 than for FXR and this should at least be discussed with appropriate references.

Reduced taurocholic acid could be the result of decreased dietary fat absorption, which would be consistent with some of the metabolic phenotypes observed. Alternatively, Defa26 increases the abundance of several microbes, some of these may be metabolising the TCA into DCA, causing a decrease in absorption and circulating TCA. A recent preprint demonstrated that many microbial taxa including Akkermansia, have the capacity to deconjugate TCA (PMID: 39868271). Given the adiposity and metabolic phenotypes associated with lipid absorption are mild and inconsistent between mouse models, we believe the second explanation is more likely. This has been included in the text.

Lines 333-338 now reads: “In both C57BL/6J and A/J mice, Defa26 supplementation lowered circulating taurocholic acid (TCA), with a more pronounced effect in C57BL6/J mice (Figure 6H). Although, TCA is a primary bile acid in mice, reductions in circulating levels sampled from the periphery (tail vein) likely reflect microbial rather than hepatic metabolism as only 5% of bile acids escape enterohepatic metabolism.”

Lines 411-417 now read: “Increased abundance may also explain some of the observed changes in circulating bile acids. Consistent with this, it has recently been shown (62) that *A. muciniphila* can metabolise up to 80% of available TCA. In both A/J and C57BL6/J mice, Defa26 supplementation appeared to decrease circulating TCA levels. While this could be the result of altered liver synthesis, as TCA is a primary bile acid in mice, it could also be the result of increased deconjugation of TCA by *A. muciniphila*.”

With respect to changes in known secondary bile acid producers, there is limited evidence that *Erysipelatoclostridium* can produce DCA (PMID: 38261437). However, we appreciate we do not have metagenomic data indicating changes in bai operon expression or BSH activity and have softened the text.

Lines 432-434 now read: “Further work is needed to validate these data as none of the microbial taxa depleted by Defa26 in A/J mice have been causally linked to DCA or TDCA metabolism.”

We have emphasised the greater affinity for Tgr5 over FXR throughout the manuscript by focussing on Tgr5 over FXR.

Lines 424-430 now read: “Both DCA and TDCA are potent agonists of Tgr5, a bile acid receptor expressed in peripheral tissues including skeletal muscle and pancreatic beta-cells. Mice lacking Tgr5 exhibit muscle wasting (56, 63), and islets treated with Tgr5 agonists exhibit increased insulin secretion (64). Decreased DCA and TDCA in response to Defa26 may inhibit bile acid signalling in these tissues, resulting in muscle wasting and hypoinsulinemia respectively.”

Lines 340-344: “Notably, both DCA and TDCA are derived from microbially mediated deconjugation reactions and are potent agonists for takeda-G-protein-receptor-5 (Tgr5; Gpbar1), a bile acid receptor expressed in skeletal muscle and beta-cells where it regulates hypertrophy and insulin secretion respectively.”

7. The data resource table or the data available statement should be provided.

Source data has been included for all main figures, and other data is available upon reasonable request.

8. The statistical tests used for each panel should be reported in every figure legend. Is there nothing significant in Figures 3B, 3C, 4A, 4I and 5J?

Main effects of one-way ANOVA have been added to figures 3B and 3C. Significance has also been added to figure 4A. Figure 4M (previously Figure 4I) and Figure 5L (previously Figure 5J) present all significantly changing microbes based on ANCOM-BC. This along with all other statistical tests have been added to the figure legends.

Minor comments:

1. There are 670 mice in total, but the AA genotype mice is 657 and AB genotype mice is 14, then it is 671 mice in total.

Genetic mapping was performed on 670 mice, we erroneously included the column header in the count. However, upon closer examination genotypes at this loci could only be accurately estimated for 452 mice including 12 AB mice, and this has been corrected. We previously subtracted the AB mice from the AA mice, and did not realise there were some mice missing genotype information. It is important to note that, missing genotype information is the result of low confidence imputation of SNP probabilities in R/qt12 and not differences in actual mouse numbers.

2. Is Figure S2A referring to Defa5 or Defa26 as indicated in the figure?

Figure S2A (now S3A) is Defa5. This has been corrected.

3. For the Figure 6D, is this bacterium also significantly changed in A/J?

Yes, the relative abundance in A/J mice is much lower than B6. We have broken the axis to clearly demonstrate this.

4. In line 431, AB DOz mice were not supplemented with alpha-defensin 26 in this study and the level of alpha-defensin 26 in the AB DOz was also not shown.

This statement has been amended. Line 411-412 now reads: “and correlated with increased expression of the antimicrobial peptide Defa26 in inbred founder strains”

5. Essential details in the Methods sections are missing and should be added, for instance, reference of the diets used, concentration of the peptides admixed to the diet, feeding status, and time of collection of the plasma for the bile acid analysis.

Feeding status and diet information can be found on lines 456-464: “All mice were maintained at 23°C on a 12-hour light/dark cycle (0600-1800) and given ad libitum access to a standard laboratory chow diet containing 16% calories from fat, 61% calories from carbohydrates, and 23 % calories from protein or an in-house high-fat high-sugar diet (western diet; WD) containing 45% calories from fat, 36% calories from carbohydrate and 19% calories from protein (3.5%g cellulose, 4.5%g bran, 13%g cornstarch, 21%g sucrose, 16.5%g casein, 3.4%g gelatine, 2.6%g safflower oil, 18.6%g lard, 1.2%g AIN-93 vitamin mix (MP Biomedicals), 4.95%g AIN-93 mineral mix (MP Biomedicals), 0.36%g choline and 0.3%g L-cysteine).”

Information on peptide concentrations can be found on lines 684-686: “Synthetic peptides were mixed into mouse diet by hand at a final concentration of 10 ng/g based on previous work (29).”

Information on bile acid sample collection can be found on line 719-720: “For bile acid quantification, plasma was collected between 1400-1600 hours.”

Dear Dr James, dear Dr Masson,

Thank you for submitting your revised manuscript (EMBOJ-2024-119542R) to The EMBO Journal, as well for your patience with our feedback. Your amended study was sent back to the referees for their scientific reassessment, and we have received re-reports from both of them, which I enclose below. As you will see, the referees state that the work has been substantially enhanced by the revisions and they are now broadly in favour of publication.

Thus, we are pleased to inform you that your manuscript has been accepted in principle for publication in The EMBO Journal.

We now need you to take care of a number of issues related to formatting and data presentation as detailed below, which should be addressed at re-submission.

Please contact me at any time if you have additional questions related to below points.

As you might remember from previous experience, every paper at the EMBO Journal now includes a 'Synopsis', displayed on the html and freely accessible to all readers. The synopsis includes a 'model' figure as well as 2-5 one-short-sentence bullet points that summarize the article. I would appreciate if you could provide this figure and the bullet points.

Thank you for giving us the chance to consider your manuscript for The EMBO Journal. I look forward to your final revision.

Again, please contact me at any time if you need any help or have further questions.

Best regards,

Daniel Klimmeck

>> Please add up to five keywords to your study.

>> Adjust the title of the 'Competing Interests' section to 'Disclosure and Competing Interests Statement'.

>> Section order should be corrected as follows: title page with complete author information, abstract, keywords, introduction, results, discussion, methods, data availability section, acknowledgements, disclosure and competing interests statement, references, main figure legends, tables, expanded figure legends.

>> References: adjust reference format to EMBO Journal format, 10 authors et al, and place References after the disclosure and competing interests statement, before figure legends.

>> Please recheck the reference for the bioRxiv entry Lucas et al. (2025) and update the citation if in the meantime published as regular article.

>> Figure callouts: Please ensure that the figures and panels are called out in the main text in sequential order. Currently, callouts for Fig. 2I-J, and Fig. 4M are missing.

>> Funding: please include the "Funding" information in the "Acknowledgements" section.

>> Reagents and Tools table for the Methods section: please as a separate .doc file using the existing template in the Guide For Authors, listing key reagents, experimental models, software and relevant equipment.

>> Rename the 'Data analysis' section to 'Statistical analysis'.

>> Table 1 should be renamed to Table EV1 with the corresponding callout; the legend should be included above the table in Excel file instead of a separate tab; An Excel file should be uploaded individually, not in a zip folder as README file is not necessary.

>> Appendix file with ToC: the title page should contain "Appendix for + manuscript title" and a ToC with the page numbers for the listed items; nomenclature should be Appendix Figure Sx throughout the manuscript and Appendix PDF; figure legends should be placed below the appropriate figures.

>> Data availability section: please include a Data availability section including a sentence stating: 'No large-scale data amenable to database repository deposition were generated in this study.'

>> Source data: Source data files need to be saved in a scheme one figure/folder and then uploaded as .zip files. E.g. all the Source data files for figure 1 need to be saved in a single folder and this needs to be zipped and then uploaded as "SD figure 1.zip" file.

>> Author Checklist: please complete the checklist entering information on: 'Cell materials>>cell line authentication'.

>> Consider additional changes and comments from our production team as indicated below:

- Figure Legends (main + EV): 1. Please note that the exact p values are not provided in the legends of figures 2C, G, H, I, J; 3B, C, E, G; 4A, B, G, H, I, L; 5C, D, G, H, J; 6C, D, E; S1 A, B, E, S3A, B, E; S4 B, S5 C, D.

2. Please note that information related to n is missing in the legends of figures 4A, 5A, S1 A

3. Please note that the error bars are not defined in the legends of figures 2B, C; 5A, F, I; S2 A, B, C, D, E, F, G; S3 A, D

Referee #1:

All of my comments have been sufficiently addressed and I would recommend acceptance of the manuscript. I congratulate the authors on this exciting study

Referee #2:

The authors have addressed my previous comments satisfactorily, and I have no further questions.

The authors addressed the remaining editorial issues.

Dear David, dear Stewart,

Thank you for submitting the revised version of your manuscript. I have now evaluated your amended manuscript and concluded that the remaining minor concerns have been sufficiently addressed.

I am thus pleased to inform you that your manuscript has been accepted for publication in the EMBO Journal.

On a different note, I would like to alert you that EMBO Press offers a format for a video-synopsis of work published with us, which essentially is a short, author-generated film explaining the core findings in hand drawings, and, as we believe, can be very useful to increase visibility of the work. Please see the following link for representative examples and their integration into the article web page:

<https://www.embopress.org/doi/full/10.15252/emj.2019103932>

Best regards,

Daniel

Daniel Klimmeck, PhD
Senior Editor
The EMBO Journal
EMBO
Postfach 1022-40
Meyerhofstrasse 1
D-69117 Heidelberg
contact@embojournal.org
